# Re-designing Interleukin-12 to enhance its safety and potential as an anti-tumor immunotherapeutic agent

Pengju Wang[1], Xiaozhu Li[2], Jiwei Wang[1], Dongling Gao[1], Yuenan Li[1], Haoze Li[1], Yongchao Chu[1], Zhongxian Zhang[1], Hongtao Liu[1], Guozhong Jiang[1], Zhenguo Cheng[1], Shengdian Wang[2], Jianzeng Dong[3], Baisui Feng[1,4], Louisa S. Chard[5], Nicholas R. Lemoine[1,5] & Yaohe Wang [1,5]

Interleukin-12 (IL-12) has emerged as one of the most potent agents for anti-tumor immunotherapy. However, potentially lethal toxicity associated with systemic administration of IL-12 precludes its clinical application. Here we redesign the molecule in such a way that its anti-tumor efficacy is not compromised, but toxic effects are eliminated. Deletion of the N-terminal signal peptide of IL-12 can effect such a change by preventing IL-12 secretion from cells. We use a newly designed tumor-targeted oncolytic adenovirus (Ad-TD) to deliver non-secreting (ns) IL-12 to tumor cells and examine the therapeutic and toxic effects in Syrian hamster models of pancreatic cancer (PaCa). Strikingly, intraperitoneal delivery of Ad-TD-nsIL-12 significantly enhanced survival of animals with orthotopic PaCa and cured peritoneally disseminated PaCa with no toxic side effects, in contrast to the treatment with Ad-TD expressing unmodified IL-12. These findings offer renewed hope for development of IL-12-based treatments for cancer.

---

[1] Sino-British Research Centre for Molecular Oncology, National Centre for International Research in Cell and Gene Therapy, School of Basic Medical Sciences, Academy of Medical Sciences, Zhengzhou University, 450052 Zhengzhou, China. [2] CAS Key Laboratory of Infection and Immunity, Institute of Biophysics, Chinese Academy of Sciences, 100101 Beijing, China. [3] Department of Cardiology, The First Affiliated Hospital of Zhengzhou University, 450052 Zhengzhou, China. [4] Section of Digestive Diseases, The Second Affiliated Hospital of Zhengzhou University, 450014 Zhengzhou, China. [5] Centre for Molecular Oncology, Barts Cancer Institute, Queen Mary University of London, London EC1M 6BQ, UK. Correspondence and requests for materials should be addressed to Y.W. (email: yaohe.wang@qmul.ac.uk)

Tumor-induced immune suppression is recognized as an important mechanism by which tumors evade immune-mediated detection and destruction[1]. A number of strategies to overcome this suppression have been evaluated, but local IL-12 expression consistently appears to be one of the most effective methods to achieve this due to its central role in T- and NK-cell-mediated inflammatory responses[2–5]. Unfortunately, clinical application of IL-12-based therapies remains problematic due to the potential for rapid development of lethal inflammatory syndrome[6–10]. The development of strategies to overcome IL-12-mediated toxicity is currently the subject of intense research and a number of modifications to IL-12 have been explored. Most recently, tumor-targeted oncolytic adenoviral (AdV) delivery of membrane-anchored IL-12 variants was analyzed in the context of efficacy against metastatic pancreatic cancer[11, 12]. However, delivery of therapeutically effective doses of AdV resulted in membrane saturation of IL-12, leading to release into the serum and subsequent toxicity. More promising drug-inducible IL-12 systems allow easier management of IL-12 levels over long periods, resulting in a reasonable degree of clinical efficacy. However, inefficient transduction of tumor cells with carrier vectors and the lack of simultaneous induction of inflammation currently limits the overall anti-tumor effect of this approach[11, 13].

Tumor-targeted oncolytic viruses (TOVs) are attractive therapeutic candidates for cancer treatment due to their ability to replicate in and directly lyse tumor cells, release tumor antigens from destroyed cancer cells and importantly induce local inflammation, which contributes significantly to reversal of local immune suppression and development of anti-tumor immune responses[14, 15]. Furthermore, TOVs can be used to efficiently deliver therapeutic genes specifically to the tumor site at an increasing level following viral replication in tumor cells. The first-generation, tumor-targeted oncolytic adenovirus, an E1B55k-deleted oncolytic adenovirus (H101) was the first OV therapy to be licensed for cancer treatment. However, although clinical safety profiles were encouraging, few objective responses were seen[16, 17]. It has subsequently been recognized that deletions in the E1B55K and E3 gene regions in the virus had a significant impact on the ability of these viruses to replicate efficiently within cells[18]. Based on our improved knowledge of AdV biology[18–20], we have constructed a new-generation replicating AdV with triple gene deletions (E1A CR2, E1B19K, and E3gp19K), Ad-TD-LUC. This was used to deliver a modified IL-12 (nsIL-12, with deletion of the IL-12 signal peptide) to Syrian hamster models of pancreatic cancer (PaCa), which are particularly suitable for these investigations as they are permissive for AdV replication[21, 22] and as shown here for the first time, permissive for human IL-12 functions.

Oncolytic viruses encoding IL-12 have demonstrated strong anti-tumor effects in preclinical models of cancers[23–25]; however, systemic accumulation of IL-12 after delivery by oncolytic viruses remains potentially lethal to patients[10, 26]. Here we report that systemic delivery of the modified nsIL-12 using our adenovirus Ad-TD-nsIL-12 to peritoneally disseminated and orthotopic pancreatic tumors is an extremely effective anti-tumor therapy. Importantly, no toxic side effects are observed, even when viruses are administered at high doses that are usually associated with lethal IL-12-mediated toxicity in these models.

## Results

**Ad-TD replicates selectively in cancer cells**. Following a better understanding of the functions of different adenovirus genes, we have constructed a novel tumor-targeted replicating AdV, Ad-TD-LUC, in which the E1ACR2, E1B19Kand E3gp19K genes were deleted and the luciferase (LUC) open reading frame inserted into the E3gp19K region (Fig. 1a). To analyze viral selectivity and replication in tumor cells, we assessed viral replication in a panel of normal and tumor cell lines (Fig. 1b–k). Ad-TD-LUC replicated efficiently in all cancer cell lines examined (Fig. 1d–k), yet was significantly attenuated in normal cell lines (Fig. 1b, c) in comparison to the wild-type Ad5 virus, which replicated to high titers in these cell lines. Furthermore, Ad-TD-LUC also demonstrated a superior in vivo selectivity. Tumor tissues, lung, and liver were examined after intraperitoneal administration of the virus into Syrian hamster bearing Hap-T1 orthotopic pancreatic cancer. While the viral E1A gene produced by Ad5 could be detected in both tumor and normal tissues at all time-points examined, E1A detected after Ad-TD-LUC treatment was confined to tumor tissue and absent from normal lung and liver tissues (Fig. 1l).

**Ad-TD-nsIL12 replicates in and is cytotoxic to cancer cells**. Oncolytic viruses encoding IL-12 have demonstrated strong anti-tumor effects in preclinical models of cancers[23–25]; however, even direct intratumoral (i.t.) administration of these agents can result in systemic accumulation of IL-12, which has the potential to trigger lethal inflammatory syndrome. To investigate the potential of IL-12 modified to prevent secretion from infected cells, Ad-TD viruses expressing wild-type IL-12 (Ad-TD-IL-12) or modified IL-12 (Ad-TD-nsIL-12) were constructed (Fig. 1a). The potency of IL-12-armed viruses, control virus and wild-type Ad5 was assessed in a panel of human cancer cell lines (Fig. 2a) and a panel of Syrian hamster tumor cell lines (Fig. 2b). The three mutant viruses were significantly more potent than Ad5 in all cell lines examined. As expected from previous data regarding the ability of oncolytic adenoviruses to replicate efficiently in Syrian hamster models[22], we found that Ad5 and our mutant viruses could replicate in Syrian hamster tumor cell lines effectively and at an equivalent level over a time course (Fig. 2c–g).

**Modifying IL-12 does not affect its biological activity**. Having confirmed that the three viruses developed in this study were able to selectively replicate in and kill tumor cells, we next investigated the intracellular accumulation and secretion of human IL-12 after infection of Syrian hamster tumor cell lines with Ad-TD-IL-12 and Ad-TD-nsIL-12. As shown in Fig. 3, while high levels of IL-12 were detected from supernatants of Ad-TD-IL-12-infected cells, very low levels of IL-12 were detected from supernatants of Ad-TD-nsIL-12-infected cells (Fig. 3a–f). IL-12 was, however, detected in lysates of Ad-TD-nsIL-12-infected cells (Fig. 3g, h), although at significantly lower levels when compared to the Ad-TD-IL-12-infected cells. The lower levels of IL-12 detected from Ad-TD-nsIL-12-infected lysates corresponded with a decrease in IL-12 mRNA detected from infected Hap-T1 cells at both 72 and 96 hpi compared to IL-12 mRNA detected in Ad-TD-IL-12-infected cells (Fig. 3h, i). No differences in mRNA levels were detected at early time points, suggesting a possibility that intracellular accumulation of IL-12 may act to downregulate its own expression at the mRNA level once an intracellular threshold is reached.

Given that human IL-12 is non-functional in murine systems, next we investigated whether human IL-12 could function within the Syrian hamster immune system. Here we show, for the first time, that recombinant human IL-12 is able to induce TNF-α (Fig. 3j) and IFN-γ (Fig. 3k) expression in hamster splenocytes ex vivo and induce proliferation of Syrian hamster, but not murine, peripheral blood mononuclear cells (PBMCs) (Fig. 3l), confirming that human IL-12 is functionally active in Syrian

hamster models and deletion of the signal peptide has no detrimental impact on its activity.

**Ad-TD-nsIL-12 is effective in a subcutaneous PaCa model.** To investigate the anti-tumor efficacy of the Ad-TD-LUC virus and IL-12-armed Ad-TDs, subcutaneous HPD1NR pancreatic tumors were established in hamsters. Animals were treated i.t. six times with $1 \times 10^9$ PFU/injection of Ad-TD-IL-12, Ad-TD-nsIL-12, Ad-TD-LUC or PBS. Multiple virus administration was used based on previous clinical studies and our recent study that demonstrates that repeated injection of oncolytic adenovirus can result in improved anti-tumor efficacy through engagement of the host

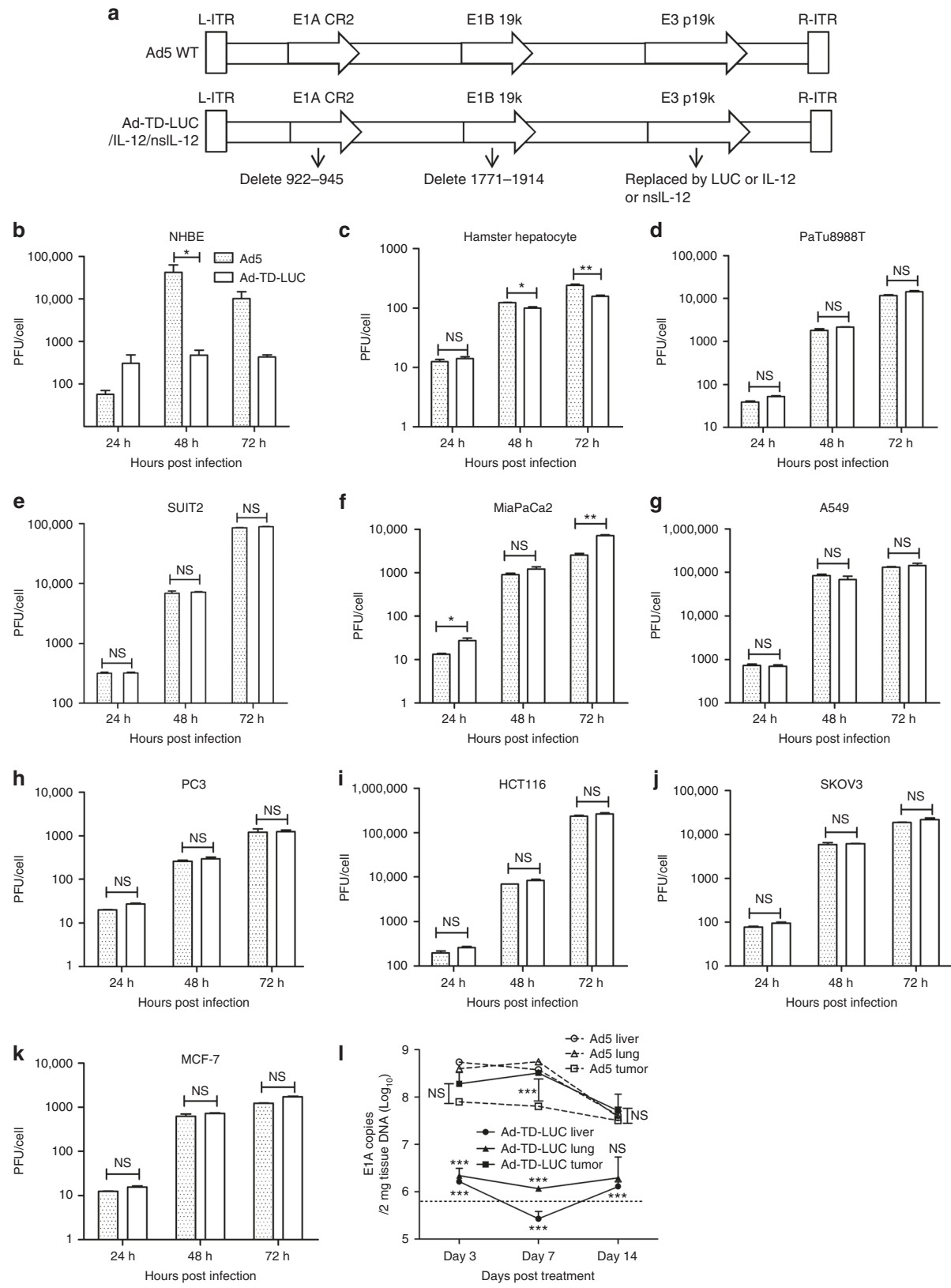

immune system against both adenovirus and tumor-associated antigens[27].

In this model, confinement of IL-12 expression to the tumor mass limited the associated toxicity resulting in 100% tumor eradication and survival of both Ad-TD-IL-12 and Ad-TD-nsIL-12-treated animals (Fig. 4a, b). Successful oncolytic virus strategies aim not only to eradicate the primary tumor, but also to induce long-term anti-tumor immunity. To evaluate the development of anti-tumor immunity for protection against disease recurrence elicited by Ad-TD-nsIL-12 treatment, those animals that had cleared their primary tumor after treatment with Ad-TD-nsIL-12 were re-challenged four weeks later with HPD1NR or control renal cancer HaK cells. Ad-TD-nsIl-12-treated animals demonstrated development of immunity to HPD1NR tumor cells, as evidenced by rapid clearance of these, but not HaK cells nor HPD1NR co-injected with hamster α-CD3 depletion antibody (Fig. 4c).

**Ad-TD-nsIL-12 can safely cure disseminated PaCa.** To evaluate the anti-tumor activity of Ad-TD-nsIL-12 in advanced PaCa, a peritoneally disseminated PaCa model that rapidly recapitulates (4–6 days) late-stage human PaCa[28] was established in Syrian hamsters (Supplementary Fig. 1a). SHPC6 cells were injected i.p. and 4 days later, PBS or virus at a dose of $1 \times 10^9$ PFU was administered i.p. on days 0, 2, and 4. Ad-TD-nsIL-12 treatment resulted in 100% survival which persisted until termination of the experiment (Fig. 5a). No animal displayed signs of treatment-related toxicity. Treatment with Ad-TD-IL-12 resulted in only a 10% survival rate, with deaths occurring earlier than those in the PBS-treated group (Fig. 5a). A dose reduction of Ad-TD-IL-12 did not alleviate IL-12-mediated toxicity, resulting in early death of a large proportion of the animals treated and no improvement in overall efficacy compared to the PBS treatment (Fig. 5b). To evaluate the improvement in safety associated with administration of Ad-TD-nsIL-12 compared to Ad-TD-IL-12, liver function was examined after PBS, Ad-TD-IL-12 ($5 \times 10^8$ PFU) or Ad-TD-nsIL-12 ($1 \times 10^9$) were administered i.p. into hamsters bearing peritoneally disseminated SHPC6 PaCa. Liver toxicity was assessed by measuring aspartate aminotransferase (AST), alanine aminotransferase (ALT) and alkaline phosphatase (ALP) levels in the serum on days 1, 3, and 5 post-injection. Significant elevations in all three enzymes were detected in the Ad-TD-IL-12-treated group at each time point whereas liver enzymes measured in Ad-TD-nsIL-12-treated animals, which received double the dose of virus, remained equivalent to those detected in PBS-treated animals (Fig. 5c). As expected, serum levels of IL-12 were higher after treatment with Ad-TD-IL-12 compared to Ad-TD-nsIL-12 levels, which remained constant throughout the time-points analyzed (Fig. 5d).

A further experiment was carried out in which one i.p. injection of high dose ($3 \times 10^9$ PFU) Ad-TD-nsIL-12 was administered. No treatment-related deaths were observed and the treated animals were completely cured, surviving until termination of the experiment (Fig. 5e). Once again, induction of long-term, tumor-specific immunity was observed after re-challenge of hamsters previously treated with Ad-TD-nsIL-12 in this model (Fig. 5f).

**Ad-TD-nsIL-12 shows efficacy in an orthotopic PaCa model.** Ad-TD-nsIL-12 was evaluated further using an orthotopic PaCa model to mimic a clinically unresectable disease scenario. The hamster HapT1 PaCa cell line was effective for establishing sizeable, metastasizing orthotopic tumors by day 6 post-injection (Supplementary Fig. 1b, c)[29]. HapT1-tumors established in the tail of the pancreas were treated i.p with PBS, Ad-TD-IL-12 or Ad-TD-nsIL-12 ($1 \times 10^9$ PFU/injection) every other day for six doses. In this model, treatment with Ad-TD-nsIL-12 significantly improved survival compared with PBS and Ad-TD-IL-12-treated animals, the latter group experiencing rapid mortality due to IL-12-mediated toxicity (Fig. 6a).

This model was used to assess an increased dose of Ad-TD-nsIL-12 ($2.5 \times 10^9$ PFU/injection, six times) vs. a reduced dose of Ad-TD-IL-12 ($5 \times 10^8$ PFU/injection, six times). Treatment with a higher dose of Ad-TD-nsIL-12 dramatically increased the survival rate compared to PBS, Ad-TD-LUC and Ad-TD-IL-12-treated animals. Reduction of the Ad-TD-IL-12 dose administered alleviated toxic effects, but efficacy remained significantly lower than that following treatment with Ad-TD-nsIL-12 (Fig. 6b). At these doses (Ad-TD-IL-12 at $5 \times 10^8$ PFU/injection and Ad-TD-nsIL-12 at $2.5 \times 10^9$ PFU/injection delivered six times i.p) we saw evidence of tumor cell infection by all viruses by immunohistochemical detection of viral proteins in the tumor sections at day 7 following the last injection (Fig. 6c) and by the titration of infectious virions recovered (Fig. 6d), which in all cases were absent from tumor samples by day 14. Analysis of viral E1A DNA produced by Ad-TD-nsIL-12 in tumor or normal tissues using this model further supported the tumor specificity of the nsIL-12-armed adenovirus at a dose of $2.5 \times 10^9$ PFU/injection, with E1A being detected at much lower levels (close to the limit of assay sensitivity) in normal lung or liver tissue compared to tumor tissue (Fig. 6e). Analysis of the growth of orthotopic Hap-T1 tumors following this regime demonstrated that both Ad-TD-IL12 and Ad-TD-nsIL-12 caused dramatic regression of the tumor size by 14 days compared to PBS and control Ad-TD-LUC (Fig. 6f), and both viruses, at their respective doses, enhanced CD3$^+$ T-cell infiltration within tumor tissue (Fig. 6g). As observed using the SHPC6 model of disseminated PaCa, systemic IL-12 accumulation was detected for the duration of viral infection only after treatment with Ad-TD-IL12 (Fig. 6h), despite both viruses being detected at comparable levels in tumor cells (Fig. 6c, d). Moreover, pathological examination of hepatocytes after treatment with Ad-TD-LUC and Ad-TD-nsIL-

**Fig. 1** A novel oncolytic adenovirus Ad-TD-LUC is tumor-selective in vitro and in vivo. **a** The basic structures of the Ad5 genome and derived mutants are shown. The arrows indicate the deleted regions in the genome of adenovirus. IL-12 comprises p40 subunit and p35 subunit linked by an elastic peptide (VPGVGVPGVG), with a signal peptide linked to the p40 subunit. Non-secreting (ns) IL-12 lacks this signal peptide. **b–k** Tumor selectivity was confirmed by assessing replication of Ad-TD-LUC and wild type AdV in normal cells and in a panel of human cancer cells. Cell lines examined were normal human bronchial epithelial cells (NHBE); primary culture hamster hepatocytes; pancreatic cancer cell lines (PaTu8988T; SUIT2; MiaPaCa2); lung cancer (A549); prostate cancer (PC3); colorectal cancer (HCT116); ovarian cancer (SKOV3) and breast cancer (MCF-7). NHBE Cells were infected at an MOI of 100 particles/cell, the rest were infected at an MOI of 5 PFU/cell. Replication assays were carried out over a period of 72 h. Infectious virus production was assessed by titration on JH293 cells and the titer as PFU/cell calculated and shown as mean and standard error of the mean (SEM). Statistical analysis was performed using a two-way ANOVA test with Bonferroni post-test; *$p < 0.05$, **$p < 0.01$; ***$p < 0.001$. Experiments were performed in triplicate. **l** Orthotopic PaCa tumors were established in Syrian hamsters using $3 \times 10^6$ Hap-T1 cells. Six days later Ad5 or Ad-TD-LUC were injected i.p. ($n = 9$/group) on day 0, 2, 4, 6, 8, and 10 at a dose of $2.5 \times 10^9$ PFU/injection. Animals were killed on day 3, 7, and 14 after the last viral treatments. Tumors, lung, and livers were analyzed by qPCR for copy numbers of the viral E1A gene. The sensitivity of the assay is illustrated by the dotted line. Mean and SEM are shown for each group and compared using an independent $t$-test. ***$p < 0.001$

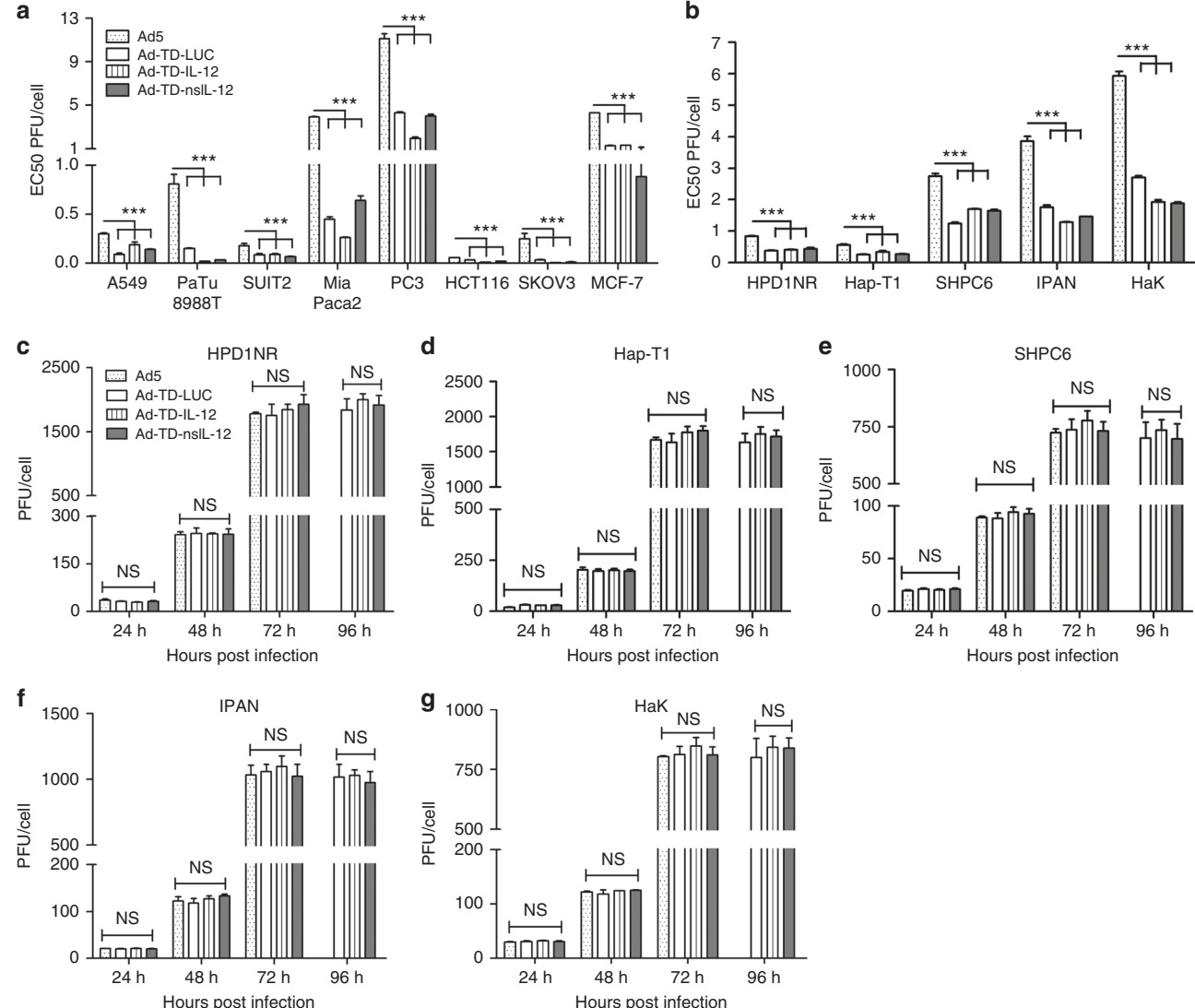

**Fig. 2** Ad-TD-LUC and IL-12-armed viruses replicate in and are cytotoxic cancer cell lines. **a**, **b** To assess virus cytotoxicity, cell proliferation assays (Promega) were carried out in all human **a** and Syrian hamster **b** tumor cell lines. Cells were infected with Ad5, Ad-TD-LUC or Ad-TD-IL-12 or Ad-TD-nsIL-12 and incubated for 6 days at 37 °C. The amount of virus particles required to kill 50% of the cells is shown as EC50 (PFU/cell) and shown as mean and SEM compared using an independent *t*-test. **c–g** Replication assays were carried out in hamster pancreatic cancer cell lines, HPD1NR, Hap-T1, SHPC6 and IPAN, and hamster kidney cancer cells (HaK) at an MOI of 5 PFU/cell. Mean and SEM are shown for each group and compared using an independent *t*-test. $*p < 0.05$, $**p < 0.01$, $***p < 0.001$

12 demonstrated only mild blood vessel congestion and eosinophilic degeneration, while Ad-TD-IL12 induced severe blood vessel congestion, eosinophilic degeneration, apoptosis and necrosis of hepatocytes (Fig. 6i), despite the dose of Ad-TD-IL-12 administered being five times lower than those of Ad-TD-nsIL-12 or Ad-TD-LUC. A dose escalation study of Ad-TD-nsIL-12 suggested that $2.5 \times 10^9$ was the optimal dose for delivery as animals receiving higher doses of Ad-TD-nsIL-12 did not show statistically improved responses to treatment demonstrating no advantage of administration of increased doses of virus (Supplementary Fig. 2a). At the highest dose examined, $2 \times 10^{10}$ PFU/injection, the survival advantage conferred by Ad-TD-nsIL-12 was actually abrogated, although serum levels of IL-12 at this dose remained low (Supplementary Fig. 2b). However, both Ad-TD-LUC and Ad-TD-nsIL-12 treatments at a such high does resulted in severe hepatotoxicity, showing severe blood vessel congestion as well as eosinophilic degeneration, apoptosis and necrosis of hepatocytes (Supplementary Fig. 2c), suggesting that the reduced survival of animals receiving the highest dose of Ad-

TD-nsIL-12 ($2 \times 10^{10}$ PFU/injection) may be derived from viral toxicity and not IL-12 related toxicity.

**Ad-TD-nsIL-12 efficacy is CD8+ T-cell dependent**. Antibodies against Syrian hamster immune cells are limited, thus to investigate further the contribution of immune subsets to treatment efficacy, we developed an α-hamster CD3+ depletion antibody[30]. Functional hamster α-CD8+ antibodies do not currently exist, thus in this instance, CD3+ CD4− cells are taken to be, in the majority, CD8+ populations. Depletion of CD3+, but not CD4+ subsets in HPD1NR subcutaneous tumor-bearing hamsters had a significant detrimental impact on treatment efficacy suggesting that Ad-TD-nsIL-12 acts via a CD3+ CD4− (CD8+) mechanism to eliminate tumors (Fig. 7a, b). Despite this, analysis of CD3+ and CD4+ infiltration into subcutaneously implanted tumors using IHC shows a significant increase in both these populations after treatment with both Ad-TD-IL-12 and Ad-TD-nsIL-12 (Fig. 7c–f). CD3+/CD4+ populations were examined further in local lymph nodes and spleens of hamsters bearing HPD1NR

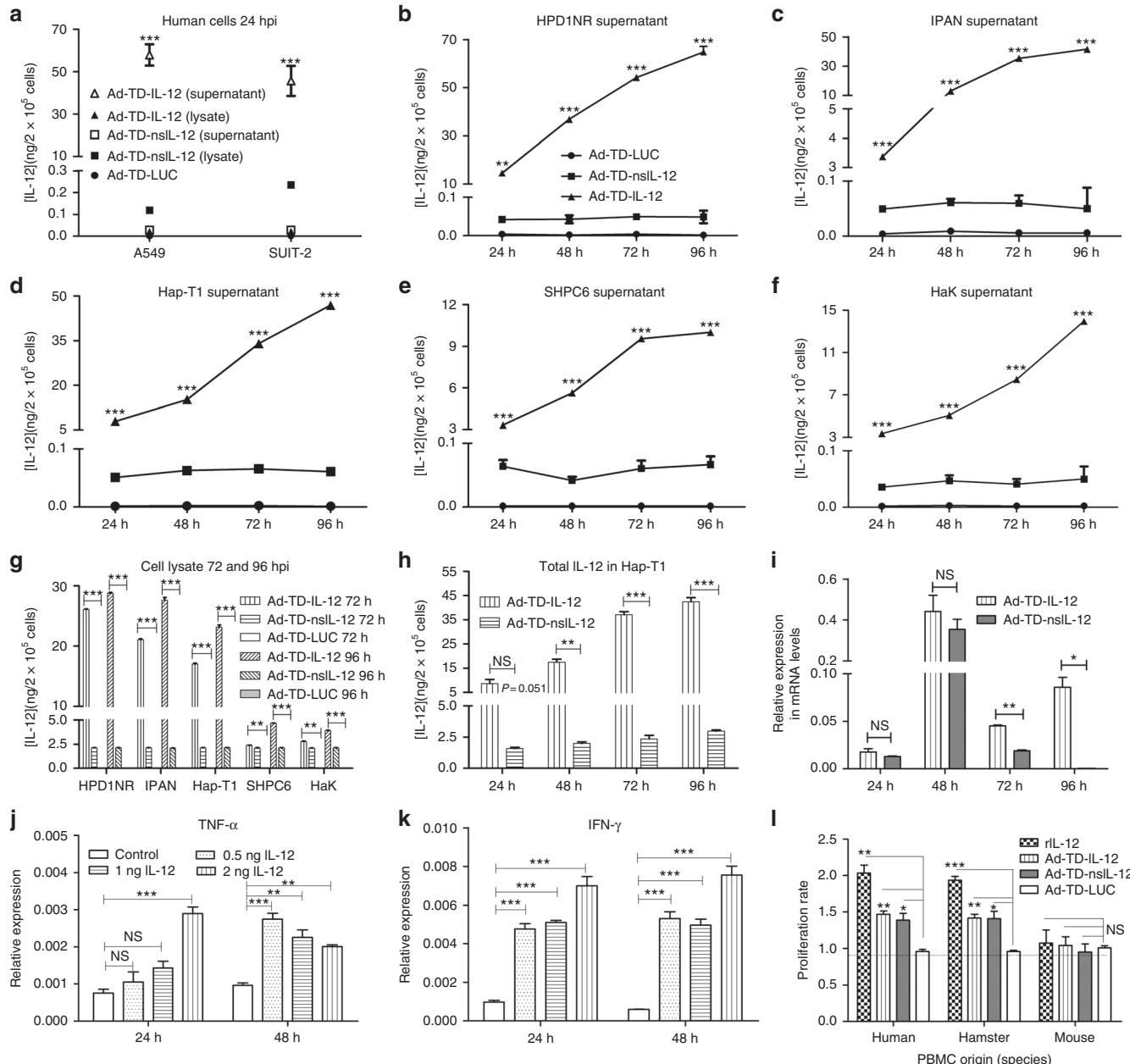

**Fig. 3** Ad-TD-nsIL-12 produced IL-12 is functionally active in Syrian hamster cells. **a–h** To detect IL-12 expression from Ad-TD-IL-12/nsIL-12 in vitro, tumor cells were infected with Ad-TD- IL-12/nsIL-12 or Ad-TD-LUC at an MOI of 5 PFU/cell. Human cancer cells **a** or Syrian hamster tumor cell supernatants **b–f** or disrupted cell lysates **g** were collected every 24 h for 96 h and assayed for IL-12 by ELISA **h** Intracellular levels of IL-12 were detected in HapT1 cells infected with Ad-TD-IL-12 or Ad-TD-nsIL-12 at an MOI of 5 PFU/cell **i** mRNA levels of IL-12 were detected in HapT1 cells infected with Ad-TD-IL-12 or Ad-TD-nsIL-12 at an MOI of 5 PFU/cell. mRNA levels relative to β-actin are shown. **j**, **k** Recombinant human IL-12 protein (rhIL-12) induced TNF-α **j** and IFN-γ **k** expression in hamster splenocytes. mRNA levels relative to β-actin are shown. **l** Human, hamster, and mouse PBMCs were incubated in the presence of (rhIL-12) or lysate from Hap-T1 cells infected with Ad-TD-IL-12 or Ad-TD-nsIL-12, each at 2 ng/ml, or Ad-TD-LUC infected lysate as a control. Cell proliferation was measured using MTS assay 48 h post-treatment. All experiments were performed in triplicate and the mean and standard error of the mean are shown for each group and compared using an independent $t$-test. *$p < 0.05$, **$p < 0.01$, ***$p < 0.001$

subcutaneous tumors treated once i.t with PBS or virus ($1 \times 10^9$ PFU) (Fig. 7g). At all time-points, treatment with Ad-TD-IL-12, but not Ad-TD-nsIL-12 resulted in significant increases in CD3+ CD4– (CD8+) populations in the spleen. Ad-TD-nsIL-12 treatment only resulted in a transient increase in CD3+ CD4+ and CD3+ CD4– (CD8+) populations in local draining lymph nodes but splenic numbers were unchanged compared to the control virus, suggesting that anti-tumor efficacy of nsIL-12 treatment depends more on alteration of T-cell function than alterations of total numbers of circulating lymphocytes (Fig. 7g).

**Ad-TD-nsIL-12 reduces expression of inflammatory cytokines.** To examine inflammatory responses further, hamster-specific qPCR primers were developed in our laboratory for analysis of inflammatory mediators (Table 1). IL-12 effects are mediated largely through IFN-γ and indeed, tumor IFN-γ levels were elevated by both Ad-TD-IL-12 and Ad-TD-nsIL-12 compared to treatment with Ad-TD-LUC or PBS. However, Ad-TD-IL-12 elevated local levels earlier and to a significantly higher degree than Ad-TD-nsIL-12 (Fig. 8). This pattern was mirrored in the detection of other inflammatory cytokines, specifically IL-2, IL-

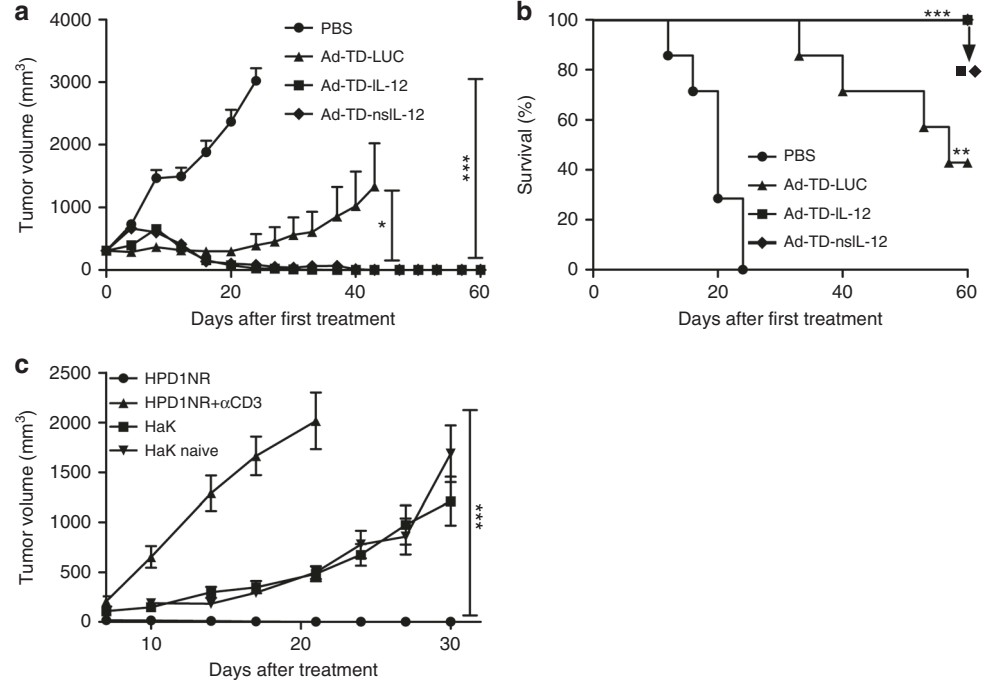

**Fig. 4** Ad-TD-nsIL-12 is an effective treatment for subcutaneous hamster PaCa. $2 \times 10^6$ HPD1NR cells were seeded into the right flank of Syrian hamsters. When tumor volumes reached 300 mm³, seven hamsters per group were each injected i.t. with 100 µl PBS, $1 \times 10^9$ PFU Ad-TD-LUC or Ad-TD-nsIL-12/IL-12 on days 0, 2, 4, 6, 8, and 10. **a** Mean tumor sizes and SEM are shown for each group and compared using a one-way ANOVA with post hoc Tukey's Multiple Comparison Test. $*p < 0.05$, $***p < 0.001$. **b** Kaplan–Meier survival curves were generated and a log-rank (Mantel–Cox) test used to analyze significance. $**p < 0.01$, $***p < 0.001$. Treatment with both Ad-TD-IL-12 and Ad-TD-nsIL-12 resulted in 100% survival in this model. **c** Hamsters that had cleared tumors after i.t treatment with Ad-TD-nsIL-12 during efficacy experiments were re-challenged 4 weeks later in the opposite flank with $4 \times 10^6$ HPD1NR or $5 \times 10^6$ HaK cells, or were injected i.p with anti-CD3 mAb (500 µg/injection) on the day before re-challenge with HPD1NR cells. In parallel, naive hamsters were injected with $5 \times 10^6$ HaK cells at the same sites and tumor growth measured and analyzed using a one-way ANOVA with post hoc Tukey's Multiple Comparison Test. $***p < 0.001$

21, IFN-β, IL-12, and IP-10, an IFN-γ-induced chemokine[31]. Thus, IL-12 release from tumor cells lysed by Ad-TD-nsIL-12 was able to stimulate effective, but reduced inflammatory responses compared to the more sustained release of IL-12 from tumor cells infected with Ad-TD-IL-12. Splenic IFN-γ and IP-10 and lymph node IFN-γ were also produced at lower levels after treatment with Ad-TD-nsIL-12 compared to Ad-TD-IL-12 and resolution occurred more rapidly following nsIL-12 treatment. These observations may account for the reduction in toxicity associated with delivery of the modified nsIL-12 by Ad-TD, compared to delivery of wild-type IL-12. Tumor CD83 expression suggested that both viruses were able to recruit mature dendritic cells with equal efficiency. Interestingly, IL-6 levels generated in response to nsIL-12 were not elevated compared to PBS or Ad-TD-LUC treatment, whereas Ad-TD-IL-12 treatment resulted in significantly enhanced levels. Elevated IL-6 levels in patients with pancreatic cancer correlate with poor survival[32], possibly due to E-cadherin down-regulation increasing metastasis[33]. Tumor IL-10 levels were also elevated more significantly after treatment with Ad-TD-IL-12 compared to Ad-TD-nsIL-12 or Ad-TD-LUC.

## Discussion

Tumor-targeted oncolytic viruses specifically infect and lyse tumor cells, release tumor-associated antigens (TAAs) and induce anti-tumor immune responses. Adenoviruses have efficacy, safety, and manufacturing characteristics that make them attractive as oncolytic virus candidates and suitable gene-delivery vectors[34]. Promising clinical data of oncolytic adenovirus *dl*1520 have shown both its anti-tumor potential and safety[35]. However,

mutant adenoviruses with E1B-55K gene deletions are markedly attenuated as oncolytic agents as E1B-55K is required for modulation of viral and cellular mRNA nuclear transport, stimulating late viral mRNA translation[36]. Subsequent generations of oncolytic adenoviruses contain rational gene deletions based on an improved understanding of the viral gene functions, however, the majority of oncolytic adenoviruses used clinically still have low potency due to E3B region deletions that result in rapid viral clearance and/or decreased viral activity[18, 37]. In this study, we created a new generation of oncolytic adenoviruses, "Ad-TD", with three deleted adenovirus genes. This virus is selective for tumor cells through the deletion of E1A conserved region 2 (CR2) and E1B-19K[38]. The third deletion, E3gp-19K has previously been shown to result in an increase in the anti-tumor effects of the virus[18]. We found that deletion of these genes improved adenovirus tumor selectivity in vitro and in vivo (Figs 1, 2) and these viruses displayed improved anti-tumor efficacy compared to first generation of oncolytic adenovirus *dl*1520[39]. In addition, therapeutic gene expression is driven by the endogenous promoter of E3-gp19k, which ensures stable and high-level gene expression during viral replication in tumor cells. Of note, the Ad-TD adenovirus retains the E3B genes.

The potent anti-tumor properties of IL-12 were first observed more than 20 years ago[40]. The multiple anti-tumor mechanisms associated with IL-12 have been intensively investigated, but clinical application remains elusive due to its short half-life and systemic toxicity[41]. Local and consistent delivery of IL-12 into the tumor microenvironment to achieve durable, low-level IL-12 expression may resolve these issues. To improve the safety and durability of IL-12 expression, we modified IL-12 by removing

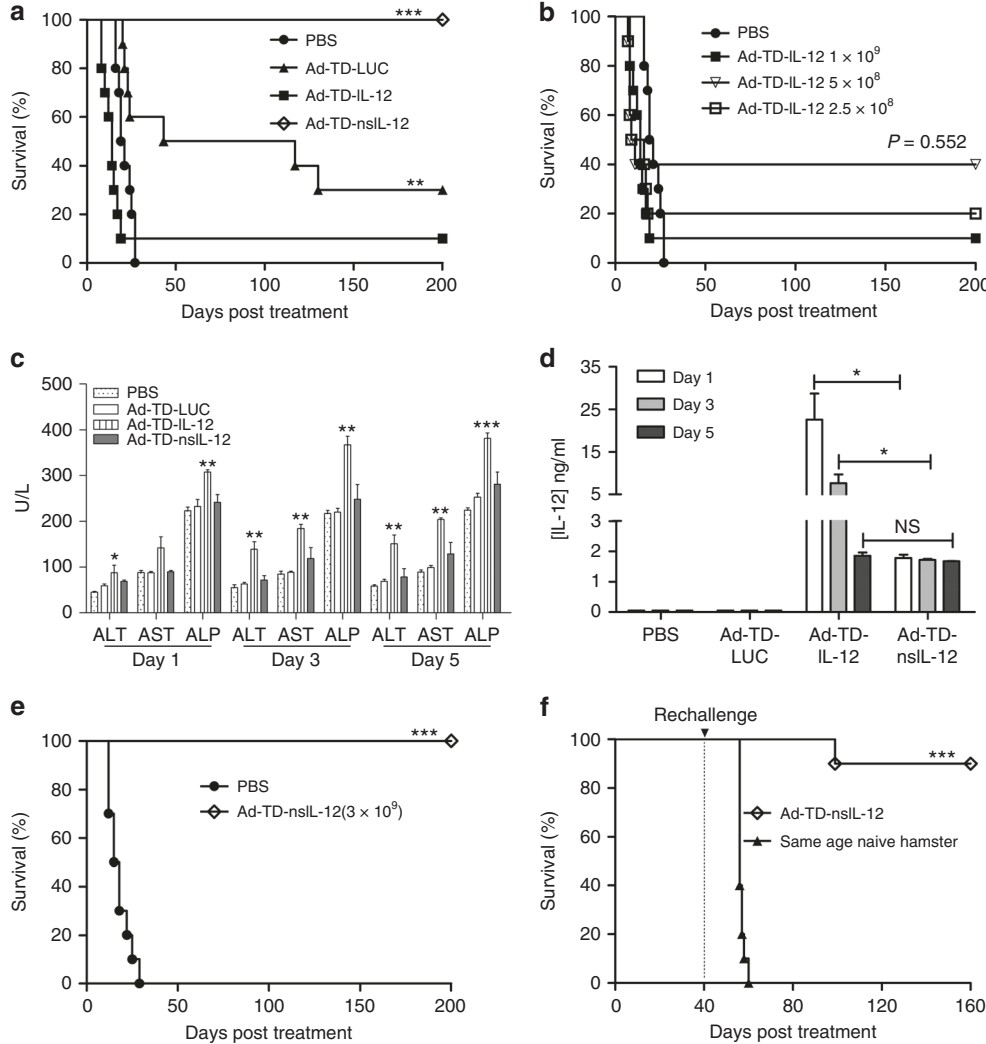

**Fig. 5** Ad-TD-nsIL-12 is a safe and effective treatment for peritoneally disseminated PaCa. $1 \times 10^7$ SHPC6 cells were seeded into the peritoneal cavity of Syrian hamsters. Four days later, 10 hamsters per group were each injected i.p with PBS or viruses. **a** PBS or $1 \times 10^9$ PFU each virus was injected i.p on days 0, 2, and 4 and survival monitored. **b** PBS or varying doses of Ad-TD-IL-12 ($2.5 \times 10^8$ to $1 \times 10^9$) were injected i.p. on days 1, 2, and 4 and survival monitored. **c**, **d** Using the same tumor model, nine hamsters per group were each injected i.p with 500 µl PBS, $1 \times 10^9$ PFU Ad-TD-LUC, $5 \times 10^8$ PFU Ad-TD-IL-12 or $1 \times 10^9$ PFU Ad-TD-nsIL-12 on day 0. Serum was collected on days 1, 3, and 5 for detection of the levels of ALT, AST, and ASP, respectively, **c** or IL-12 **d** detected by ELISA. Mean and SEM are shown. Statistical analysis was carried out using a one-way ANOVA with post hoc Tukey's Multiple Comparison Test or independent *t*-test. *$p < 0.05$, **$p < 0.01$, ***$p < 0.001$. **e** $3 \times 10^9$ PFU Ad-TD-nsIL-12 or PBS was administered i.p on day 0 and survival monitored ($n = 10$). **f** PBS or $1 \times 10^9$ PFU Ad-TD-nsIL-12 was injected i.p on days 0, 2 and 4 ($n = 10$). Forty days later, surviving hamsters previously treated with Ad-TD-nsIL-12 were re-challenged with $1 \times 10^7$ SHPC6 cells. Age-matched naive hamsters also were peritoneally injected with SHPC6 cells as control. Survival was monitored. In each case, Kaplan–Meier survival curves were generated and significance assessed using the log-rank (Mantel–Cox) test. **$p < 0.01$, ***$p < 0.001$

**Fig. 6** Ad-TD-nsIL-12 is an effective and safe treatment for orthotopic PaCa in Syrian hamsters. Six days after Hap-T1 cells were seeded into the tail of the pancreas, hamsters were injected i.p. with 500 µl PBS, $1 \times 10^9$ PFU Ad-TD-LUC, Ad-TD-IL-12 or Ad-TDnsIL-12 (**a**) or $2.5 \times 10^9$ PFU Ad-TD-LUC, $2.5 \times 10^9$ PFU Ad-TD-nsIL-12 or $5 \times 10^8$ PFU Ad-TD-IL-12 (**b**) ($n = 7$/group) on days 0, 2, 4, 6, 8, and 10. Kaplan–Meier survival curves were generated. Significance was assessed using the log-rank (Mantel–Cox) test. *$p < 0.05$, **$p < 0.01$. **c–h** Animals were treated as **b** ($n = 3$/time point/group) and killed on day 3, 7, and 14 after last viral treatments. **c** Representative images of immunohistochemical staining for Hexon at day 7. Hexon-positive cells were counted in five high-power fields (HPF) from each tumor section (×200). ND: not detected. **d** Infectious virion recovery from tumor tissue was determined by TICD50 using JH293 cells. **e** Tumors, lung, and livers were analyzed by qPCR for the copy numbers of the viral E1A gene after treatment with Ad-TD-nsIL-12 at $2.5 \times 10^9$ PFU/injection on day 0, 2, and 4. The sensitivity of the assay is illustrated by the dotted line. Mean and SEM are shown for each group and compared using an independent *t*-test. *$p < 0.05$, **$p < 0.01$. **f** Mean tumor volumes and SEM are shown for each group. Statistical analysis was carried out using a one-way ANOVA with post hoc Tukey's Multiple Comparison Test. *$p < 0.05$. **g** Representative images of immunohistochemical staining for CD3 at day 7. Quantitative scores of lymphocyte infiltration within tumors are shown (right panel) from five HPF from each tumor section (×200). The scoring was conducted within the tumor and stroma and necrotic areas were avoided. The extent of positive cells was categorized into the following four grades: 1, <15 cells/HPF; 2, 16–30 cells/HPF; 3, 31–45 cells/HPF; 4, >45 cells/HPF. ND: not detected. **h** IL-12 expression in sera was detected by ELISA. **i** Representative histopathology of HE staining of livers after three i.p injections of virus into hamsters bearing orthotopic PaCa tumors ($n = 3$/group) on day 0, 2, and 4, using the same dose as in **b**. Livers were collected one day following the last injection and analyzed using HE staining (×200)

the signal peptide required for secretion from cells and used our adenovirus AD-TD to deliver this directly or systemically to a number of different tumor models. When Ad-TD-nsIL-12 was injected i.p into hamsters bearing intraperitoneally disseminated SHPC6 PaCa, only low levels of IL-12 were detected in peripheral blood on day 1, 3, 5, and no obvious changes of ALT, AST and ALP liver enzymes were detected (Fig. 5). Delivery of unmodified

IL-12 resulted in high levels of peripheral IL-12 and elevations in liver enzyme levels, consistent with severe liver blood vessel congestion, eosinophilic degeneration and hepatocyte apoptosis and necrosis (Fig. 6), indicative of induction of systemic toxicity. By preventing IL-12 secretion from tumor cells, nsIL-12, tumor-associated antigens and new Ad-TD-nsIL-12 virions are released only when the tumor cells were lysed after viral infection, limiting

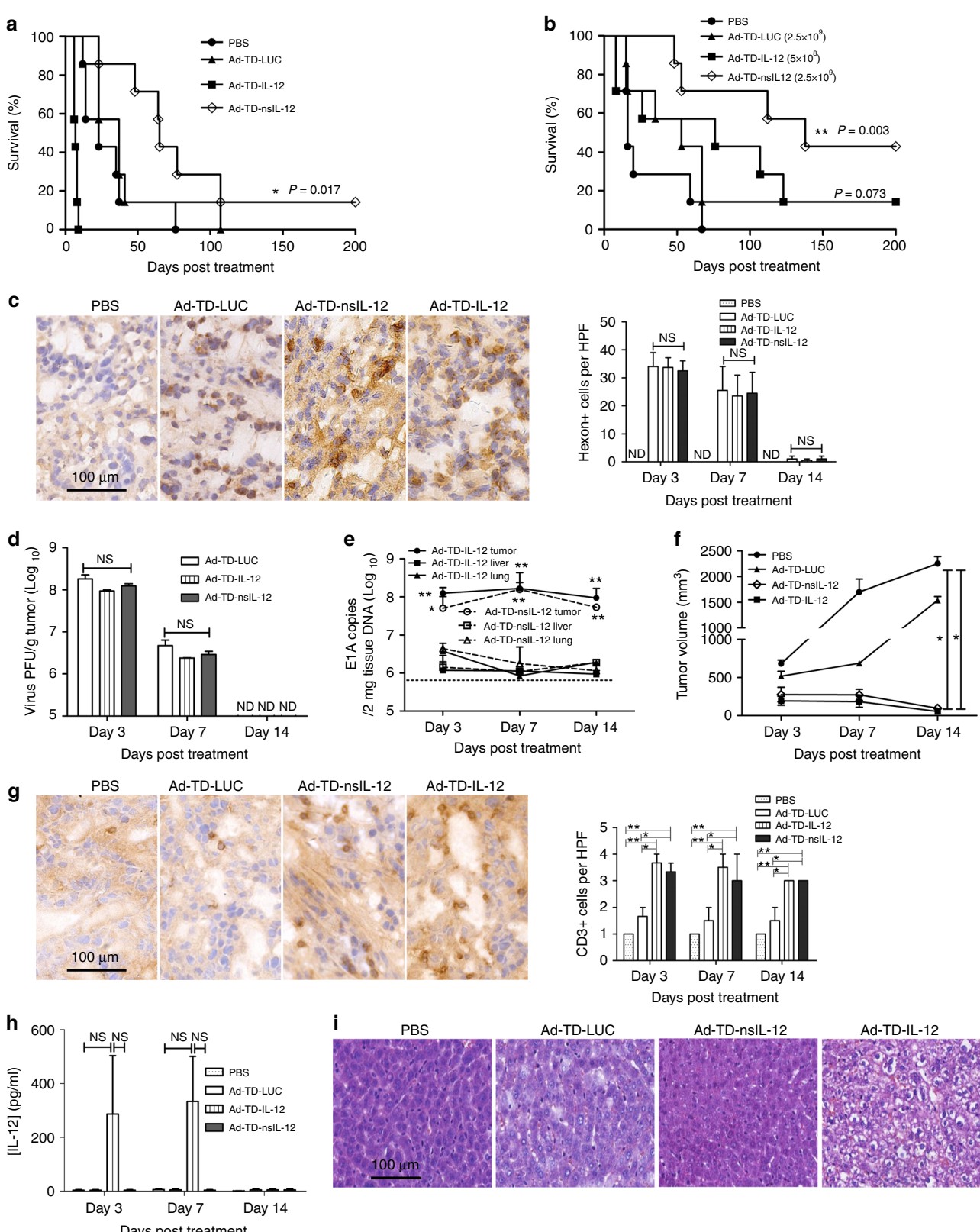

the dissemination of IL-12 to the local tumor environment where effective anti-tumor immunity was elicited (Fig. 6).

It has been reported that human IL-12 does not stimulate mouse PBMC proliferation[42], but no study has been published assessing the effect of human IL-12 on hamster PBMCs. PBMC proliferation assays presented here demonstrate that human IL-12 stimulates both human and hamster PBMCs proliferation, but not mouse PBMCs. In addition, rhIL-12 was able to effectively stimulate IFNγ and TNFα expression by activated splenocytes ex vivo (Fig. 3). These results suggest Syrian hamsters as an

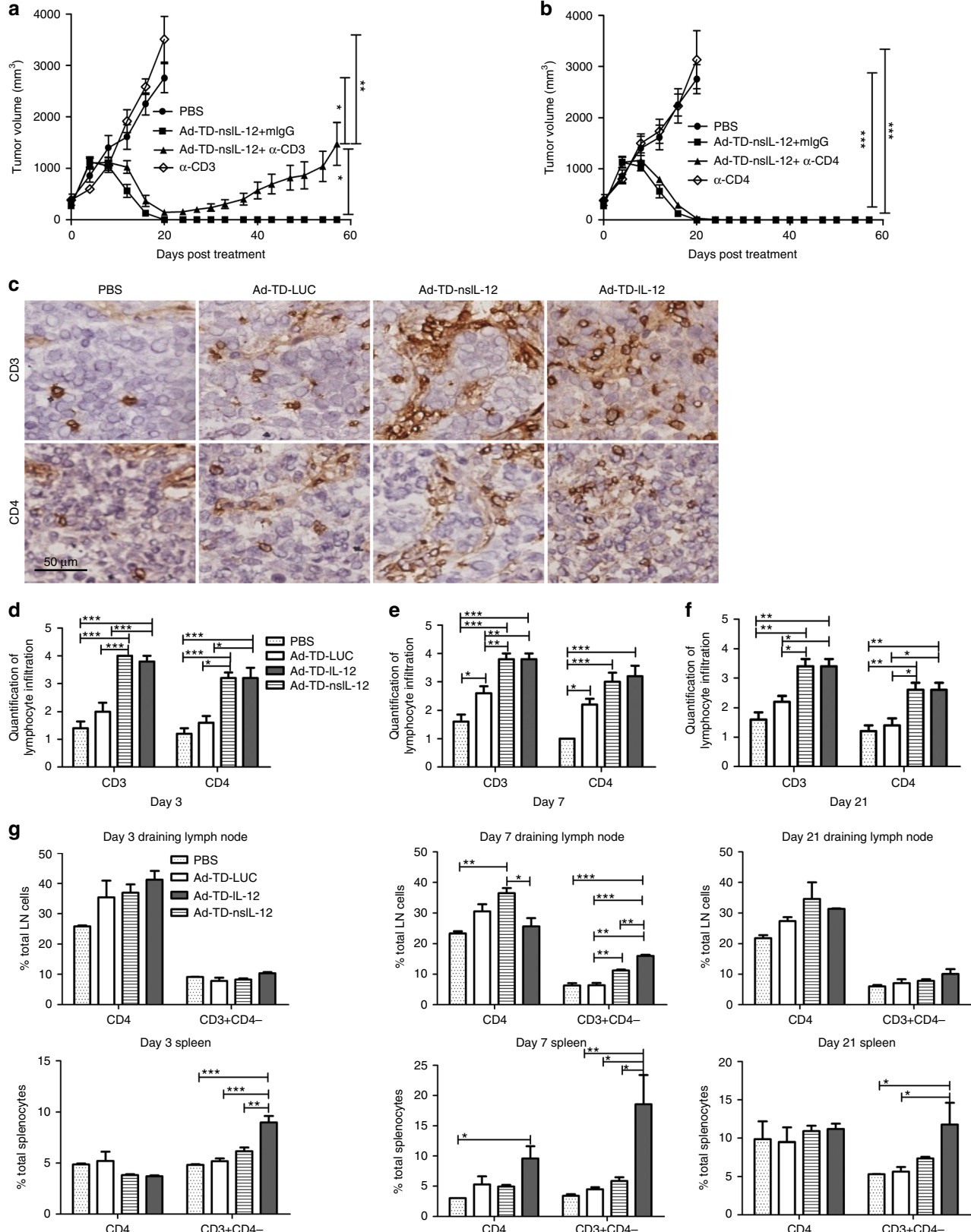

effective model for assessing the anti-tumor effects of both oncolytic adenoviruses and human inflammatory cytokines.

Given previous reports of the ability of IL-12 to promote T-cell proliferation and program effector functions within the stable population of human CD8+ effector memory T cells in vivo[43, 44], we examined the reliance of our treatment on these immune subsets. Ad-TD-nsIL-12 therapy generated robust anti-tumor memory T cells as evidenced by rejection of secondary tumors. The lack of suitable research tools for analysis of Syrian hamster immune subsets has limited the scope of research possible in this model, but we have developed and validated a number of Syrian hamster-specific antibodies and qPCR primers that allow initial investigation of the immune responses elicited by our viruses. Depletion of CD4+ T cells had no effect on treatment efficacy, whereas depletion of CD3+ cell populations had a negative impact on treatment efficacy in vivo (Fig. 7). Taken together these results suggest an importance of CD3+ C4− (CD8+) cells for treatment efficacy.

Further investigation of immune functions, using FACS analysis and qPCR detection methods developed in our laboratory revealed that Ad-TD-nsIL-12 therapy had no effect on splenic T cells and only transiently increased the percentages of CD4+ and CD8+ (CD3+ CD4−) populations in local draining lymph nodes compared with the control virus, suggesting that anti-tumor efficacy of nsIL-12 treatment depends more on modification of T-cell function than alterations of total numbers of circulating lymphocytes (Fig. 7), although currently the lack of tools available for analysis of T-cell activation status in Syrian hamster models precludes confirmation of this.

IL-12 effects are mediated largely through IFN-γ and indeed, tumor IFN-γ levels were elevated by both Ad-TD-IL-12 and Ad-TD-nsIL-12 compared to treatment with Ad-TD-LUC or PBS (Fig. 8). However, Ad-TD-nsIL-12 elevated local levels of IFN-γ and other inflammatory cytokines later and to a lesser extent than Ad-TD-IL-12. Splenic IFN-γ and IP-10 and lymph node IFN-γ were also produced at lower levels after treatment with Ad-TD-nsIL-12 compared to Ad-TD-IL-12. These observations suggest that the improved safety associated with nsIL-12 delivery may be due to dampening, but not abrogation of the inflammatory responses generated by IL-12 and this hypothesis warrants further investigation. Tumor IL-10 levels were also elevated more significantly after treatment with Ad-TD-IL12 compared to Ad-TD-nsIL12 or Ad-TD-LUC. A wealth of evidence now exists to suggest that IL-10 functions as both an immune-suppressive and immune-stimulating or anti-tumor cytokine, depending on the context of its expression[45–47], thus the role of IL-12-induced IL-10 in this setting requires further study.

In this study, we have developed a robust and safe IL-12 based immune-viro-therapeutic agent that can be delivered locally or systemically to tumors. Syrian hamster models of pancreatic cancer, which can accurately reflect stages of tumorigenesis in humans, were treated effectively and safely using Ad-TD-nsIL-12 using treatment regimes comparable to those currently applied clinically (Figs. 4–6). However, further assessment of the pharmacology and toxicology of this approach is still warranted prior to translation of this therapy into early stage clinical trials for pancreatic cancer. Of note, this agent might be also effective for other advanced solid tumors such as gastric, colorectal, and ovarian cancers for which the virus can be intraperitoneally injected. In summary, the approach developed in this study offers renewed hope for development of IL-12-based treatments for cancer.

## Methods

**Study design.** The objective of this study was to investigate the functional efficacy and safety of modified IL-12 when delivered to in vivo tumor models using a new oncolytic adenovirus. All hamster procedures were approved by the Animal Welfare and Research Ethics Committee of Zhengzhou University (Zhengzhou, China). For in vivo experiments, power calculations were carried out to determine required sample sizes using G*Power 3[48], F-test ANOVA repeated measures setting parameters of $\alpha = 0.1$, power = 90%, effect size = 0.5, group = 3. In subcutaneous tumor models, animals were randomly assigned by an independent animal caretaker to treatment groups by matching tumor sizes prior to treatment. Tumor growth was measured using electronic callipers until a tumor measured 2.0 cm in diameter or ulcerated, at which point the animal was killed. Tumor growth curves were terminated upon the death of the first animal in each group, but group survival was monitored until the experimental endpoint (all animals in group deceased, or 60 days post treatment). For orthotopic and disseminated pancreatic cancer models, animals were assigned randomly to treatment groups and animal survival was monitored by assessment of animal well-being every other day. Experimental endpoints in these models were 200 days post treatment. Animal caretakers were blinded to treatment groups in all cases.

**Cell lines.** The Syrian hamster pancreatic ductal adenocarcinoma (PDAC) cell lines HPD1NR (maintained in Roswell Park Memorial Institute (RPMI) 1640 with 10% FCS), Hap-T1, SHPC6, IPAN and the renal tumor cell line HaK were maintained in Dulbecco's modified Eagle's medium (DMEM) with 10% FCS. SHPC6 and IPAN were kindly provided by W. Wold (St. Louis University, St. Louis, MO, USA). HPD1NR and Hap-T1 cell lines were purchased from the German Collection of Microorganisms and Cell Cultures. The HaK cell line was purchased from the American Type Culture Collection (ATCC). JH293, the human kidney epithelial cell line transformed with Ad5 DNA, was obtained from the Cancer Research UK Central Cell Services (London, United Kingdom) and maintained in DMEM with 10% FCS. Normal human bronchial epithelial cells (NHBE) were obtained from Cambrex (Cambridge, UK) and maintained in Bronchial Epithelial Growth Medium (BEGM) (Cambrex). Primary culture hamster hepatocytes (maintained in DMEM with 10% FCS) were isolated and cultured in our lab. All cells were maintained at 37 °C in a humidified atmosphere containing 5% $CO_2$ and confirmed to be mycoplasma-free before being used experimentally.

**Viruses and antibodies.** Adenovirus type 5 (Ad5) mutants were generated by homologous recombination as described previously[49]. The complete Ad5 genome was used as the backbone in all new mutants and was derived from the pTG3602 plasmid. The following viruses were generated: Ad-TD (triple deletion: E1ACR2, E1B19K, and E3gp19K-deleted), Ad-TD-LUC (luciferase, from pGL3-Basic Vector

**Fig. 7** Ad-TD-nsIL-12 efficacy is dependent on hamster CD3+/CD4− immune cell subsets. **a, b** Syrian hamsters were inoculated subcutaneously with $2 \times 10^6$ HPD1NR cells. The established tumors were injected i.t. with $1 \times 10^9$ PFU Ad-TD-nsIL-12 or PBS (n = 7/group) on day 0, 2, 4, 6, 8, and 10. Control IgG and either mouse anti-hamster CD3 mAb (4F11) **a** or CD4 mAb **b** were injected i.p. at doses of 500 μg/injection every fourth day from the day before the viral therapy to the end of the experiment and FACS analysis used to confirm the depletion. Mean tumor volumes and SEM are shown for each group. Statistical analysis was carried out using a one-way ANOVA with post hoc Tukey's Multiple Comparison Test. *$p < 0.05$, ***$p < 0.001$. **c–g** $2 \times 10^6$ HPD1NR cells were seeded into the right flank of Syrian hamsters. When tumor volumes reached 300 mm³, nine hamsters per group were each injected i.t with PBS, $1 \times 10^9$ PFU Ad-TD-LUC or Ad-TD- IL-12/nsIL-12 on day 0. On days 3, 7, and 21 tumors were collected and processed for IHC. **c** Representative images of immunohistochemical staining for CD3 and CD4 at day 7 (×200). **d–f** Quantitative scores of lymphocyte infiltration within tumors. Lymphocytes were counted in 5 HPFs randomly selected from each tumor section (×200). The scoring was conducted within the tumor and stroma; necrotic areas were avoided. The extent of lymphocyte infiltration was categorized into the following four grades: 1, <25 cells/HPF; 2, 25–49 cells/HPF; 3, 50–75 cells/HPF; 4, >75 cells/HPF. Statistical analysis was carried out using a one-way ANOVA with post hoc Tukey's Multiple Comparison Test. *$p < 0.05$, **$p < 0.01$, ***$p < 0.001$. **g** Spleens and lymph nodes were collected and analyzed by FACS for CD3 and CD4 expression at the time points shown. Mean expression and SEM is plotted (n = 3/group). Statistical analysis was carried out using a one-way ANOVA with post hoc Tukey's Multiple Comparison Test. *$p < 0.05$, **$p < 0.01$, ***$p < 0.001$

**Table 1 qPCR primers designed to amplify mRNA of Syrian hamster immune modulators**

| Gene | Forward | Reverse |
|---|---|---|
| IFN-γ | 5′-TGTTGCTCTGCCTCACTCAG-3′ | 5′-CACCAGCCTTTTGCCAGTTC-3′ |
| IP10 | 5′-GACCGACCGGTAAAACCGAG-3′ | 5′-CACGTGGGCAGGATTGACTT-3′ |
| CD83 | 5′-CCCAGAGCAGGCAAAACAAC-3′ | 5′-TTCCTGAAAGGTGACTCGGC-3′ |
| IL1b | 5′-CGTGGACCTTCCAGGATGAG-3′ | 5′-AGCTGTCGAATGGGAGCATC-3′ |
| IL-2 | 5′-CTCGCATCCTGTCTTGCACT-3′ | 5′-AGCATCATGGGGAGTTTCGG-3′ |
| IL-6 | 5′-ATAGTCACGCCTAGCCCAAC-3′ | 5′-TCTTGGTTCTTGGCCACTCC-3′ |
| IL-10 | 5′-AGTAACTGCACCCACTTCCC-3′ | 5′-TGGCAACCCAAGTAACCCTT-3′ |
| IL-12 | 5′-AGGCTCTGAATCTCAACGGC-3′ | 5′-GATTGTCACAGCACGGATGC-3′ |
| IL-15 | 5′-CGGGTCATTTTGCACGAGTA-3′ | 5′-CCTTGCAGCCATGTTCTGTT-3′ |
| IL-21 | 5′-ACGCTCAGCTTTTGCCTGTT-3′ | 5′-GCTCTTCTTCTGCCTTCTCGT-3′ |
| TNF-α | 5′-TTCTCCTTCCTGCTTGTGGC-3′ | 5′-CAGGCTTGTCGTTCGAATTTTG-3′ |
| β-actin | 5′-AGATGACCCAGATCATGTTTGAGA-3′ | 5′-CAGGATGGCATGAGGGAGAG-3′ |

(Promega), Ad-TD-IL-12 (p40 with signal peptide and p35 without signal peptide fragments were cloned from the cDNA derived from RPMI-8866 cells, linked with elastin cDNA by PCR) and nsIL-12 (non-secreted IL-12, p40, and p35 without signal peptide fragments were cloned from the former IL-12 by PCR). LUC, IL-12, and nsIL-12 genes were driven by the endogenous E3gp19k promoter[50]. The mouse monoclonal antibody (mAb) against Syrian hamster CD3e (clone 4F11, IgG1 isotype), mAb against mouse CD4 (GK1.5, cross-reacting with Syrian hamster) and anti-KLH mAb were prepared as previously described[30, 50].

**Cytotoxicity assay.** Cells were seeded at $2 \times 10^3$ to $4 \times 10^3$ cells per well in 96-well plates and infected with viruses 16–18 h later at starting MOI = 1000 PFU/cell. Cell survival on day 6 after viral infection was determined by MTS assay as detailed by the manufacturer (Promega) and EC50 values (viral dose killing 50% of tumor cells) were calculated as previously described[18]. All assays were performed at least three times.

**Viral replication assay.** Cells were seeded at $2 \times 10^5$ cells per well, in three wells of 6-well plates in medium with 10% FCS, and infected at 100 particles or 5 plaque-forming units (PFU)/cell of Ad5 or Ad-TD-LUC/IL-12/nsIL-12 16–18 h later. Primary culture hamster hepatocytes were infected immediately after isolation. Samples were collected in triplicate at 24-hour intervals up to 72 or 96 h after infection, freeze-thawed three times, and titrated on JH293 cells to determine the 50% tissue culture infective dose (TCID50) as previously described[18].

**TNF-α and IFN-γ expression induced by IL-12.** Hamster splenocytes were obtained from healthy donors. A total of $4 \times 10^7$ fresh splenocytes were seeded in a 12-well plate at a concentration of $2 \times 10^6$ cells/ml and stimulated with PHA for 3 days. The activated splenocytes were seeded in a 12-well plate at a density of $4 \times 10^6$ cells/well and incubated with 100 U/ml of human IL-2 for 24 h. The activated splenocytes were washed with PBS, then seeded in a 12-well plate at a density of $4 \times 10^6$ cells/well and incubated with human rIL-12 for 24 or 48 h at 37 °C. The mRNA levels of TNF-α and IFN-γ were detected by qPCR.

**PBMC proliferation assay.** Human, hamster, and mouse peripheral blood was obtained from healthy donors. PBMCs were separated on a Ficoll-Hypaque density gradient. A total of $8 \times 10^6$ fresh PBMCs were seeded in a 6-well plate at a concentration of $2 \times 10^6$ cells/ml and stimulated with PHA for 3 days. The activated PBMCs were seeded in a 96-well plate at a density of $1 \times 10^5$ cells/well in a final volume of 50 μl and incubated with 50 μl of the lysate of Hap-T1 cells infected with Ad-TD-IL-12 or Ad-TD-nsIL-12 for 48 h at 37 °C. PBMC proliferation was determined by MTS assay. rIL-12 (2 ng/ml) was used as a positive control. Samples were set-up in triplicate.

**ELISA.** Cell lysate, supernatant and serum samples were generated and processed as described in other sections of the methods. IL-12 levels were quantified using an IL-12-specific ELISA (eBioscience) according to the manufacturer's instructions and 1:10–200 dilutions were used, with samples tested in triplicate.

**Real-time PCR.** Fresh tissues were homogenized and total RNA extracted using Trizol (Invitrogen). cDNA was synthesized by reverse transcription (Promega). qPCR was carried out using the Bio-Rad CFX Real-Time PCR System and the SYBR Green assay (Bio-Rad) to quantify cytokine expression. Primers of Syrian hamster TNF-α, IFN-γ, IP10, CD83, IL-1β, IL-2, IL-6, IL-10, IL-12, IL-21, and β-actin genes (Table 1) were designed using Primer Premier 5.0 software (Premier) and constructed by Sigma-Aldrich. Data were analyzed using Bio-Rad CFX Manager (Bio-Rad). Tissue DNA was extracted using the DNeasy blood and tissue kit (Qiagen). qPCR was carried out using the ABI STEPONE PLUS system and the SYBR Green assay (TAKARA) to detect viral genome in tissues. The primers of

adenovirus E1A gene are E1A-Forward: 5′-TGATCGATCCACCCAGTGAC-3′ and E1A-Reverse: 5′-ATGACAAGACCTGCAACCGT-3′. Data were analyzed using ABI STEPONE PLUS system software. Results were normalized to PBS-treated group values.

**Histopathological examination and immunohistochemistry.** The tissues collected at different time points were processed and stained by H&E staining or immunohistochemistry (IHC) for CD3 and CD4 as previously described[18].

**FACS analysis.** Spleens and draining lymph nodes were extracted from hamsters, combined with complete T-cell medium (RPMI medium, 10% FCS, 1% penicillin/streptomycin, 1% sodium pyruvate) then pushed through a 70 μm cell strainer to create a single cell suspension. Cells were centrifuged and the pellet incubated in 5 ml red blood cell lysis buffer (Sigma-Aldrich). Splenocytes and lymph node cells ($1 \times 10^6$) were prepared and stained with mAb against Syrian hamster CD3 and mAb against mouse CD4-conjugated with FITC (GK1.5, cross-reactive with Syrian hamster, BD), followed by secondary APC-conjugated rabbit anti-mouse antibody for CD3 staining (BD). Cells were acquired on a FACS scanner, and data were analyzed using FlowJo software (Tree Star Inc).

**Assessment of infectious virus.** Fresh tumor tissues were homogenized and titrated on JH293 cells to determine the 50% tissue culture infective dose (TCID50) as previously described[18]. Viral DNA levels were evaluated in tumors and livers using real-time PCR as described above and previously[30].

**In vivo animal studies.** Subcutaneous tumor model: A total of $2 \times 10^6$ HPD-1NR cells were implanted subcutaneously into the right flank of female, 4-week to 5-week-old Syrian hamsters. When tumors reached 6–8 mm in diameter, hamsters were stratified into groups of seven animals to receive 50 or 100 μl intratumoral (i.t) injections of Ad-TD-LUC, Ad-TD-IL-12, Ad-TD-nsIL-12, or PBS as specified in the figure legends. The injections were introduced through a single central tumor puncture site, and three to four needle tracks were made radially from the center while virus was injected as the needle was withdrawn. Tumor volumes were estimated (volume = (length × width² × π)/6) twice weekly until tumors reached 2.0 cm in diameter or tumor ulceration occurred.

For biological time-point experiments to investigate functional mechanisms, hamsters were stratified into groups when tumors reached 6–7 mm in diameter and treated once on day 0. On day 3, 7, and 21, tumors, lymph nodes, and spleens were collected from three animals in each group to investigate lymphocyte populations, immunohistochemical staining for CD3 and CD4 expression in tumor tissues and tumor-specific immunity.

**Orthotopic pancreatic cancer model.** 4-week to 5-week-old Syrian hamsters were anaesthetized with 10% chloral hydrate by an intraperitoneal (i.p) injection. Hamsters were placed in the dorsal decubitus position, and a left subcostal incision was made. The pancreas was carefully exposed, and 50 μl of tumor cell suspension ($6 \times 10^7$ cells/ml) was injected into the tail of the pancreas with a 25-gauge needle. A technically successful injection was characterized by the formation of a visible bubble within the pancreatic parenchyma. The needle was slowly withdrawn to avoid macroscopic cell spread from the injection site. The pancreas was then returned to the peritoneal cavity, and the abdominal wall was closed in two layers with nylon sutures. Six days later, the tumor volumes reached to 4–5 mm in diameter, and seven hamsters per group were each injected i.p with 500 μl PBS, $1 \times 10^9$ PFU Ad-TD-LUC or $5 \times 10^8$ PFU Ad-TD-IL-12 or $1 \times 10^9$ PFU Ad-TD-nsIL-12 on days 0, 2, 4, 6, 8, and 10. The survival of hamsters was monitored.

**Intraperitoneally disseminated pancreatic cancer model.** A total of $1 \times 10^7$ SHPC6 cells were seeded into the lower right peritoneal cavity of Syrian hamsters.

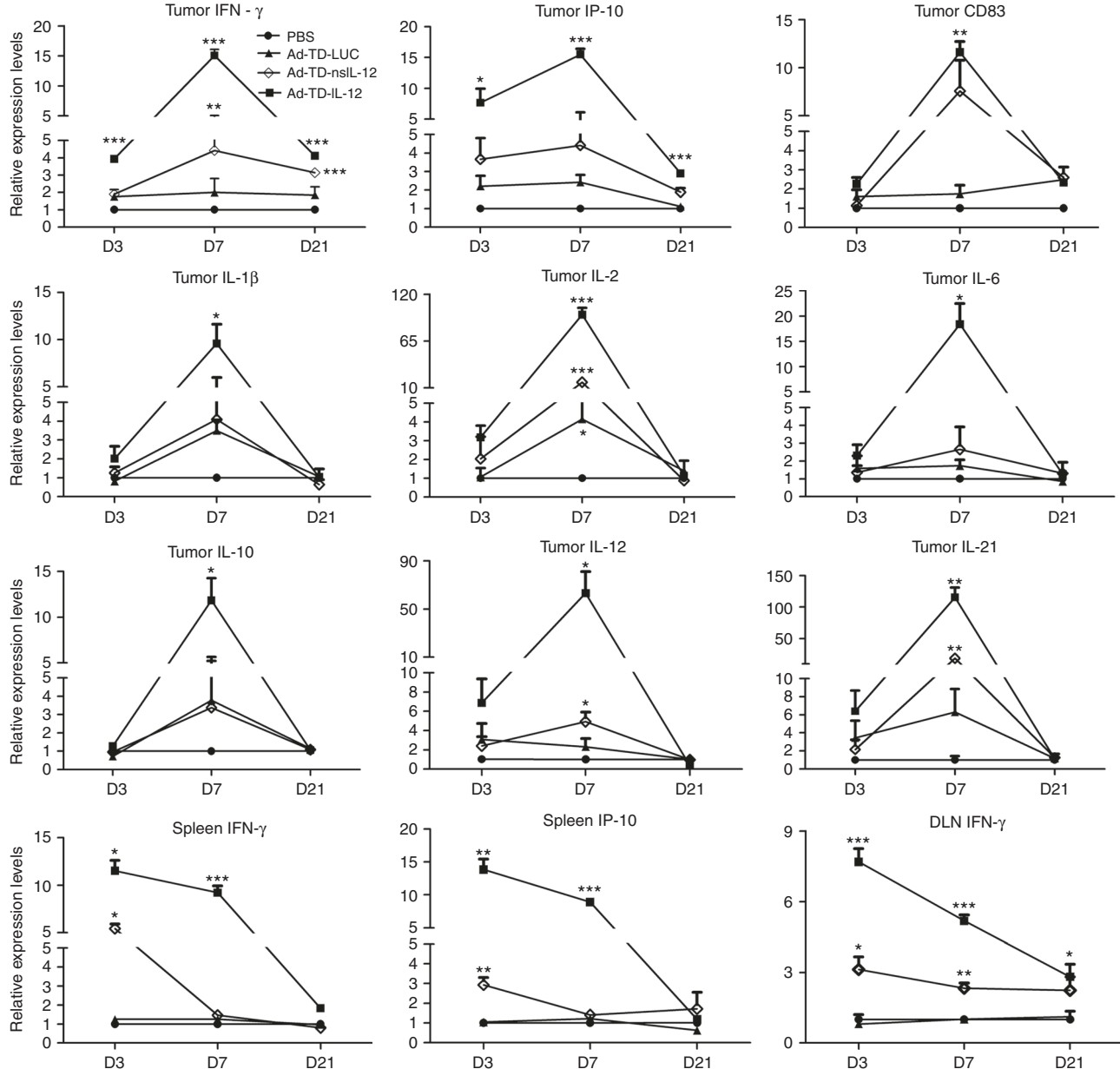

**Fig. 8** Ad-TD-nsIL-12 efficacy is mediated by production of inflammatory mediators. $2 \times 10^6$ HPD1NR cells were seeded into the right flank of Syrian hamsters. When tumor volumes reached 300 mm$^3$, nine hamsters per group were each injected i.t with 100 µl PBS, $1 \times 10^9$ PFU Ad-TD-LUC or Ad-TD- IL-12/nsIL-12 on day 0. Spleens, draining lymph nodes (DLN) and tumors were collected and analyzed by qPCR to analyze expression levels of CD83, IFN-γ, IP10, IL-1β, IL-2, IL-6, IL-10, IL-12, and IL-21 at the indicated time points. All experiments were performed in triplicate and the mean and SEM are shown for each group compared using an independent t-test. *$p < 0.05$, **$p < 0.01$, ***$p < 0.001$

Four days later, 10 hamsters per group were injected i.p with 500 µl PBS, $1 \times 10^9$ PFU Ad-TD-LUC or $5 \times 10^8$ PFU Ad-TD-IL-12 or $1 \times 10^9$ PFU Ad-TD-nsIL-12 on days 0, 2, 4. Alternative doses were given as indicated in the text or figure legend. The survival of hamsters was monitored.

**Hepatotoxicity evaluation.** Three hamsters per group per time point were injected i.p with $1 \times 10^7$ SHPC6 cells. Four days later, nine hamsters per group were each injected i.p with 500 µl PBS, $1 \times 10^9$ PFU Ad-TD-LUC or $5 \times 10^8$ PFU Ad-TD-IL-12 or $1 \times 10^9$ PFU Ad-TD-nsIL-12 on days 0. Blood was collected from hamsters from the retro-orbital sinus, and the sera were analyzed for transaminase levels and IL-12 expression on day 1, 3, 5.

**Re-challenge of tumor free animals.** The hamsters that underwent complete subcutaneous tumor regression following Ad-TD-IL-12 or Ad-TD-nsIL-12 treatment were re-challenged with $4 \times 10^6$ HPD-1NR cells (twice the number of cells compared to the primary tumor cell inoculation) or $5 \times 10^6$ HaK cells after primary tumors had not been detected for 1 month. One group of the cured animals were

injected i.p with anti-CD3 mAb on the day before the re-challenge with $4 \times 10^6$ HPD-1NR cells. Tumor volumes were measured twice weekly. SHCP6 cell re-challenge is described in the figure legend.

**CD3 and CD4 depletion in vivo.** A total of $2 \times 10^6$ HPD-1NR cells were implanted subcutaneously into 4-week to 5-week-old Syrian hamsters. When tumors reached 300 mm$^3$, hamsters were distributed between the treatment and the control groups by matched tumor size to receive i.t injections of $1 \times 10^9$ PFU Ad-TD-nsIL-12 or PBS on day 0, 2, 4, 6, 8, and 10. Depletion mAb against Syrian hamster CD3 (clone 4F11), mAb against mouse CD4 or control Ig (mouse anti-KLH mAb) were administered i.p with doses of 500 µg/injection every fourth day from the day before the virotherapy to the end of the experiment and FACS analysis used to confirm depletion. Tumor volumes were measured as in the efficacy experiments.

**Statistical analysis.** Statistical analysis was carried out using Graph Pad Prism 5 and SPSS 19.0 software. The results were represented as mean ± standard or deviation (SD) or ± standard error of the mean (SEM). Differences between

groups were analyzed using the Student's *t*-test, one-way ANOVA test or Kaplan–Meier survival analysis. Differences were considered statistically significant when the *p*-value was less than 0.05.

**Data availability**. All data presented are available from the authors on request.

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

## Acknowledgements

This project was supported by the National Natural Science Foundation of China (81272525 and 81201792), Ministry of Science and Technology of China (2013DFG32080) and The MRC (MR/M015696/1). All contributing authors have agreed to the submission of this manuscript for publication. Originating from a joint collaboration between the Sino-British Research Centre and Beijing Bio-Targeting

Therapeutics Inc. who identified an unexpected mutant Ad-TD-IL12 in which the cytokine is expressed in a non-secreting form.

## Author contributions

Y.W. supervised the whole project. Y.W. and P.W. designed the study and confirmed the mutant IL12 is non-secreted. P.W. performed the majority of the experiments; X.L. did biological time points experiments and analyzed the data. D.G. and Z.C. performed the histopathological study; J.W., Y.L., H.L. and Z.Z. contributed to animal experiments; Y.C. purified and titrated adenovirus; H.L. and G.Z. made pShuttle plasmids for the viral construction; S.W. participated in the design and interpretation of some experiments; S.W., J.D., B.F. and N.R.L. critically reviewed the paper. P.W., L.S.C. and Y.W. interpreted all results and wrote the manuscript.

## Additional information

**Competing interests:** Y.W., P.W., D.G., and N.R.L. are inventors of a filed patent application relevant to this study. The remaining authors declare no competing financial interests.

**Change history:** A correction to this article has been published and is linked from the HTML version of this paper.

