## [Peer Review File · Nature Communications]

Reviewers' comments:

Reviewer #1 (Remarks to the Author):

Oncolytic adenoviruses armed with immunostimulatory transgenes such as IL-12 have a dual mechanism of action: direct oncolytic effect and expression of the cytokine. One of the limitations of this approach is the intense secretion of IL-12 from infected cells, which reaches toxic levels at relatively low viral doses, insufficient to reach a relevant oncolytic effect. To solve this problem, the authors describe a new oncolytic adenovirus (Ad-TD-nsIL-12) encoding a modified human IL-12 protein in which the signal peptide has been deleted. Regulation of viral replication is achieved by a combination of previously described strategies (E1A-CR2 and E1b19k deletions). The transgene is inserted in substitution of the E3 p19K gene, leaving the E3B region intact. After testing the specificity of replication in cancer cells, the virus is extensively analyzed for IL-12 release and in vivo toxicity/antitumor effect in different pancreatic cancer models established in Syrian hamsters. Overall, the authors demonstrate that the new IL-12 version reduces the toxicity of the virus and improves its therapeutic window. This constitutes an advance in the field of immunovirotherapy. Experiments on the mechanism of action indicate that CD8+ T cells play a relevant role, as observed in other models.

General comments:

1. The authors use different cell lines to establish subcutaneous, intraperitoneal or intrapancreatic models, and use these models to address different aspects. They should justify this choice because such diversity of models complicates the interpretation of results. In principle, it would be more logical to use the same cell line for intraperitoneal and intrapancreatic tumors and then validate the results with a different cell lines using the same anatomical locations.
2. The development of tumors before initiation of treatment is relatively brief (4 days after cell implantation for intraperitoneal tumors, 6 days for intrapancreatic). This limits the translational relevance of results, because pancreatic cancer is usually diagnosed in an advanced stage.
3. The authors should comment on the clinical feasibility of multiple virus administrations following the schedule used in the experiments.

Specific comments:

1. Figure 1b should indicate the PFUs produced at different times in cancer and normal cells, not just the ratio. Taking into account that the ratio is maintained, and that Ad-TD-nsIL-12 should be amplified in cancer cells over time, this means that the virus is also amplified in normal cells.
2. In figure 1c, comparison with wild type adenovirus would be more relevant.
3. In figure 3, both secreted and intracellular IL-12 are expressed as ng/ml. However, the final concentration of IL-12 contained in cellular extracts will depend on the volume of resuspension. For a better comparison, IL-12 should be expressed as total ng/cell number.
4. The authors have deleted the signal peptide of IL-12 to avoid secretion from infected cells. In fact, a reduction of IL-12 in the supernatant of cells is observed. However, the reason for this reduction is not clear. Figures 3g and 3h show that cells infected with Ad-TD-nsIL-12 accumulate less intracellular IL-12 than cells infected with Ad-TD-IL-12. One possible explanation is that IL-12 devoid of signal peptide is degraded in the cell. The authors should investigate this possibility.
5. In line with the lack of local IL-12 accumulation is the fact that IFN γ in tumor extracts is also lower in the case of Ad-TD-nsIL-12 (figure 8). However, lymphocyte infiltration is not reduced. The author should discuss this apparently contradictory result.
6. As stated by the authors, the most clinically relevant model used in this work is the orthotopic implantation of the aggressive HaP-T1 cells. Not surprisingly, the antitumor effect was partial in this model, despite the increase in the viral dose (figures 6a and 6b). Is this the maximal tolerated dose for Ad-TD-nsIL-12? If toxicity limits dose escalation, is it due to high IL-12 levels in serum?

Minor comment:

The explanation of different ways to control virus replication is not relevant in the discussion section because this is not the main topic of the manuscript.

Reviewer #2 (Remarks to the Author):

Overview:

Interleukin-12 (IL-12) is a potent anti-tumor immunotherapeutic, yet its toxicities after system delivery can be severe and dose-limiting. The authors have re-designed the IL-12 molecule and expressed it from an oncolytic adenovirus with the goals of limiting systemic toxicities while maintaining its anti-cancer immunostimulatory effects. The IL-12 molecule they designed is unable to be secreted from the cell, and therefore is only released from tumor cells following cell lysis. The oncolytic adenovirus into which the transgene for nsIL-12 was engineered is a gene-deleted adenovirus with previously described modifications combined. The Syrian hamster pancreatic tumor model was utilized for studying in vivo efficacy due to previous publications suggesting that this species supports adenovirus replication and toxicity, initially within the lungs. The authors report that this approach is superior to oncolytic adenovirus expressing standard IL-12.

Methods:

The authors should address the following items:

- ♣ Activity of IL-12 in Syrian hamsters: The authors state that WBC proliferation is sufficient to demonstrate activity of human IL-12 in this species. However, IL-12 has numerous, complex and species-specific functions. This endpoint alone is not enough to state all important/relevant IL-12 functions are intact within this species.
- ♣ Efficacy data: The authors should consider treating tumors at a more advanced stage to better reflect the clinical situation. These animals were treated shortly after tumor cell implantation when tumors were not yet established and/or confirmed to be metastatic.
- ♣ Route of administration: The authors should test IV therapy given that pancreatic cancer is frequently systemic and metastasized after diagnosis. In addition, pancreatic tumors are extremely difficult to inject directly clinically given the high intratumoral pressure and fibrosis.
- ♣ Cancer cell selectivity in vitro: This is assessed in a single human lung cancer line and one normal cell type. This data on selectivity and efficacy should be expanded with more cancer lines (including pancreatic) and normal cell sources. Wildtype and gene deletion Adeno controls should also be evaluated in parallel.
- ♣ Cancer selectivity assessed in vivo: This study assessed E1A gene copies per unit tissue. However, it is possible that replication was incomplete/inefficient in hamster tissue, and therefore the infectious virus titers may be significantly lower. The dosing frequency (every two days for 10 days total) may also result in virus genomes being present despite inefficient replication. The authors should assess plaque-forming units (pfu) in addition to E1A gene copies. In addition, other normal tissues should be assessed (eg proliferating tissues which may be more supportive of viral replication). Wildtype and gene deletion Adeno controls should also be evaluated in parallel. Finally, replication and selectivity of the Ad-nsIL-12 virus should be assessed in vivo.
- ♣ Syrian hamster cell line infection and replication: These data should be shown alongside data on replication of both Adeno gene-deletion mutants (as was done), wildtype adenovirus control(s), and all viruses in human pancreatic cancer cell lines. This data is necessary to interpret the replication efficiency of these gene-deletion adenovirus constructs in hamster cell lines.

Reviewer #3 (Remarks to the Author):

In this manuscript, Wang et al. report success in treating a Syrian hamster model of pancreatic cancer with an altered oncolytic adenovirus delivering modified human IL-12. Overall, the study builds on the generally accepted idea that IL-12 delivery could improve anti-tumor responses due to many of its immune-related actions, particularly if its related toxicity could be reduced. Indeed, this is seemingly what this group has accomplished. Because of this and the fact that this group used a human IL-12, which could be quickly translated to clinical trial, this manuscript should be published. While much of the work presented is well rationalized and executed, a number of issues should be addressed before this manuscript is suitable for publication:

Major (in order of importance, with most important being listed first):

- 1) The comparison of Figure 7a with modified IL-12-treated but CD4-depleted or CD8-depleted hamsters to IL-12 treated non-depleted hamsters is not the correct control. The question remains what is the growth of tumors in untreated but CD4-depleted and CD8-depleted hamsters. What if the tumor growth curves are the same in IL-12-treated and untreated hamsters that have been CD4-depleted or CD8-depleted. Then CD4s or CD8s would seemingly not play any role (but that cannot be determined without the untreated CD4-depleted and untreated CD8-depleted controls).
- 2) The conclusions from Figure 8 are too far-reaching. There is a statement made that "these observations suggest that the improved safety associated with nsIL-12 delivery is due to reduction, but not abrogation of the inflammatory responses generate by IL-12." While I agree that reduced inflammatory responses generated by IL-12 are shown, there is no indication that this has anything to do with improve safety. These two findings may be unrelated. Some indication that putting inflammation back (via CPG ODN or poly IC) in the context of nsIL-12 reproduces the lack of efficacy of normal IL-12, would "suggest" the claim made here. Otherwise, this claim should be removed.
- 3) Inconsistent statistical analyses: some figures seemingly have no statistical analyses (including figure 2, figure 3, panels c, d, and f of figure 6, figure 8) although seemingly differences are claimed to exist among groups within those figures and their panels.
- 4) In Figure 5f, an important rechallenge experiment (40 days later) is conducted to demonstrate long-term efficacy, and the proper rechallenge control of naïve hamsters being injected is mentioned in the text. However, in the figure it seems that only such naïve hamsters were use in the original challenge and none were used after the day 40 rechallenge (to show how much different equally age-matched naïve hamsters would respond). Another (maybe even more pertinent) manner in which to ask this question is: if hamsters were challenged with this tumor, then the tumor was excised (without the mice ever receiving the modified IL-12), would they demonstrate response to the rechallenge shown in Figure 5f, or would they essentially appear naïve?

Minor:

- 1) Line 41 states that tumor-induced suppression is "the" major mechanism by which tumors evade immune-mediated detection. While it is certainly "a" mechanism, it may not be "the" mechanism since it has not been shown to be more "major" than the lack of immune infiltration (i.e., cold tumors), neoantigen sequestration from the immune system, and immune effector exhaustion (against which checkpoint inhibitors have been successful in clinical trials and as first line treatment regimens). This statement should be revised.

Responses to reviewers' comments

Reviewer #1 (Remarks to the Author):

Oncolytic adenoviruses armed with immunostimulatory transgenes such as IL-12 have a dual mechanism of action: direct oncolytic effect and expression of the cytokine. One of the limitations of this approach is the intense secretion of IL-12 from infected cells, which reaches toxic levels at relatively low viral doses, insufficient to reach a relevant oncolytic effect. To solve this problem, the authors describe a new oncolytic adenovirus (Ad-TD-nsIL-12) encoding a modified human IL-12 protein in which the signal peptide has been deleted. Regulation of viral replication is achieved by a combination of previously described strategies (E1A-CR2 and E1b19k deletions). The transgene is inserted in substitution of the E3 p19K gene, leaving the E3B region intact. After testing the specificity of replication in cancer cells, the virus is extensively analyzed for IL-12 release and in vivo toxicity/antitumor effect in different pancreatic cancer models established in Syrian hamsters. Overall, the authors demonstrate that the new IL-12 version reduces the toxicity of the virus and improves its therapeutic window. This constitutes an advance in the field of immunovirotherapy. Experiments on the mechanism of action indicate that CD8+ T cells play a relevant role, as observed in other models.

General comments:

1. The authors use different cell lines to establish subcutaneous, intraperitoneal or intrapancreatic models, and use these models to address different aspects. They should justify this choice because such diversity of models complicates the interpretation of results. In principle, it would be more logical to use the same cell line for intraperitoneal and intrapancreatic tumors and then validate the results with a different cell lines using the same anatomical locations.

Reply: We agree with the reviewer that it is better to use the same cell line for different cancer models. However, different cell lines modelling pancreatic cancer in the Syrian hamster behave in different ways according to their location, which reflect the different stages of pancreatic cancer. For example, while HPD1NR cells form subcutaneous tumours effectively and consistently, HPD1NR cells do not grow well and were not invasive when

used as an orthotopic model. However, HAP-T1 cells in this model could induce liver and lymph node metastasis (Abraham *et al*, Pancreas 2004) and some tumor cells in this model could break through the tumor capsule and enter the peritoneal cavity (Supplementary Fig 1).

SHPC6 cells grow by semi-suspension culture. Our data and those of other groups report that SHPC6 injected intra-peritoneally demonstrate neoplastic progression similar to end-stage human pancreatic cancer with malignant ascites. By 4 days after injection, the abdominal cavity contains considerable amount of serosanguinous ascites fluid and multiple oval-shaped nodules localize and grow in the mesentery adjacent to the pancreas and spleen (Supplementary Fig 1, and Spencer *et al*, Cancer Gene Ther. 2009). This model mimics tumor metastasis and invasion from the primary tumor. We have included a justification of our choice for different tumor models in our revised manuscript.

2. The development of tumors before initiation of treatment is relatively brief (4 days after cell implantation for intraperitoneal tumors, 6 days for intrapancreatic). This limits the translational relevance of results, because pancreatic cancer is usually diagnosed in an advanced stage.

Reply: In response to this comment, and to show translational relevance of our models, we show in our revised manuscript that four or six days (model-dependent) is adequate for replication of late-stage disease. At six days after intrapancreatic injection, Hap-T1 tumors reached 6-7mm in diameter, and some tiny metastatic tumors were found in the liver (Supplementary Fig. 1). At four days after intraperitoneal injection of SHPC6 cells, the abdominal cavity had considerable amounts of serosanguinous ascites fluid, multiple oval-shaped nodules had localized and grown in the mesentery adjacent to the pancreas and spleen (Supplementary Fig 1).

All of these pathological conditions show intraperitoneal and intrapancreatic tumors are in advanced stages and thus do represent the most commonly diagnosed advanced stage disease.

We have clarified this within the relevant text sections:

Ad-TD-nsIL-12 is an effective and non-toxic anti-tumor agent after systemic delivery for treatment of peritoneally disseminated PaCa

And

Ad-TD-nsIL-12 retains superior antitumor efficacy in a clinically relevant orthotopic pancreatic cancer model

We have also added supplementary Figure 1 to show images of tumours/pathology in these models after such short timeframes.

3. The authors should comment on the clinical feasibility of multiple virus administrations following the schedule used in the experiments.

Reply: A first-generation human adenovirus 5 (H101, Shanghai Sunway Biotech Co., China) has been approved as the world's first oncolytic virotherapy (Garber. J Natl Cancer Inst. 2006). Shanghai Sunway Biotech Co. owns the global patent rights of ONYX-015 (dl1520), which was the first of these viruses to be tested for human pancreatic cancer treatment. In patients, H101 has been delivered via intratumoral or intraperitoneal injection for 5 consecutive days in patients at a dose of 2×10^{11} particles. A phase II trial enrolled patients with squamous cell carcinoma of the head and neck (SCCHN) who had recurrence/relapse after prior conventional treatment. Patients received ONYX-015 at a dose of 2×10^{11} particles via intratumoral injection for either 5 consecutive days (standard) or twice daily for 2 consecutive weeks (hyperfractionated) during a 21-day cycle (Nemunaitis et al. Journal of Clinical Oncology, 2001). A phase I trial of intraperitoneal delivery of dl1520 in patients with recurrent ovarian cancer has also been assessed. Sixteen women with recurrent/refractory ovarian cancer received 35 cycles (median, two cycles) of dl1520 delivered on days 1 through 5 (Vasey et al. Journal of Clinical Oncology, 2002).

In our study, hamsters received viruses via intra-tumoral or intraperitoneal injection 6 times on day 0, 2, 4, 6, 8, 10. Our unpublished data show that this regimen is better than 5 or 6 consecutive days. The mechanism of different regimens resulting in diverse antitumor efficacy is being analysed and will be published in a later paper, however the clinical use of oncolytic adenoviruses to date suggests our regime is clinically feasible. We have briefly referenced the clinically applicability of this regime in our revised discussion.

Specific comments:

1. *Figure 1b should indicate the PFUs produced at different times in cancer and normal cells, not just the ratio. Taking into account that the ratio is maintained, and that Ad-TD-nsIL-12 should be amplified in cancer cells over time, this means that the virus is also amplified in normal cells.*

Reply: Originally, we aimed to show the selectivity of our Ad-TD vector in normal lung epithelial cells and a matched lung cancer cell line, and we presented the ratio. Following the reviewer's comment, we have revised the figure based on the comment to indicate PFUs produced in each case. Although the cells used for viral replication are normal (non-tumor), the cells are still proliferating when cultured *in vitro* using specific medium with multiple growth factors, therefore it is not surprising that the Ad-TD virus still can replicate in the "normal cells", however it is significantly attenuated compared to wild type adenovirus. When the virus was injected into live Syrian hamsters, which have an intact immune system and normal cells in key organs are largely in the quiescent phase, our Ad-TD did not replicate in normal organs such as lung and liver (see new data in Fig 1l and Fig 6 d and e).

2. *In figure 1c, comparison with wild type adenovirus would be more relevant.*

Reply: We have revised the figure based on comment to illustrate replication of wild-type adenovirus (Ad5) versus Ad-TD-LUC in all cell lines.

3. *In figure 3, both secreted and intracellular IL-12 are expressed as ng/ml. However, the final concentration of IL-12 contained in cellular extracts will depend on the volume of resuspension. For a better comparison, IL-12 should be expressed as total ng/cell number.*

Reply: We agree with this and have revised Figure 3 based on the comment and in each case, ng/2x10⁵ cells is displayed.

- 4. The authors have deleted the signal peptide of IL-12 to avoid secretion from infected cells. In fact, a reduction of IL-12 in the supernatant of cells is observed. However, the reason for this reduction is not clear. Figures 3g and 3h show that cells infected with Ad-TD-nsIL-12 accumulate less intracellular IL-12 than cells infected with Ad-TD-IL-12. One possible explanation is that IL-12 devoid of signal peptide is degraded in the cell. The authors should investigate this possibility.*

Reply: The reduction of intracellular IL-12 protein expression after infection with Ad-TD-nsIL-12 compared to Ad-TD-IL-12 is an interesting finding and a topic that warrants further investigation although it is not the key objective of this study. We investigated by qPCR the changes in mRNA-levels of IL-12 after Ad-TD-IL12 and Ad-TD-nsIL-12 infected Hap-T1 cells. These data are shown in Figure 3i and indicates that while at 24 and 48 hpi mRNA levels were equivalent, at 72 and 96 hours post-infection, intracellular IL-12 mRNA levels decreased significantly after infection with the Ad-TD-nsIL-12 virus compared to the Ad-TD-IL-12 virus. This suggests the possibility that intracellular accumulation of IL-12 can feed back to prevent further IL-12 production at the mRNA level.

- 5. In line with the lack of local IL-12 accumulation is the fact that IFN γ in tumor extracts is also lower in the case of Ad-TD-nsIL-12 (figure 8). However, lymphocyte infiltration is not reduced. The author should discuss this apparently contradictory result.*

Reply: The total number of TILs is similar, but the number of activated T cells and the activity of the infiltrating T cells may be different after Ad-TD-nsIL-12 and Ad-TD-nsIL-12 treatment. This has been clarified in the Discussion section. At present the lack of tools to investigate Syrian hamster immune cells precludes analysis of T cell activation status. We are in the process of developing methods to detect changes in T cell status in this model.

6. *As stated by the authors, the most clinically relevant model used in this work is the orthotopic implantation of the aggressive HaP-T1 cells. Not surprisingly, the antitumor effect was partial in this model, despite the increase in the viral dose (figures 6a and 6b). Is this the maximal tolerated dose for Ad-TD-nsIL-12? If toxicity limits dose escalation, is it due to high IL-12 levels in serum?*

Reply: We thank the reviewer for this insightful comment and have performed a dose escalation study to investigate this further (new Supplementary Figure 2). 2.5×10^9 PFU/injection remains the optimal regime. We now show in Supplementary Figure 2 that when the dose is increased to 5×10^9 , 1×10^{10} and 2×10^{10} PFU/injection, the survival rate of the Syrian hamsters treated at the highest doses was not increased. 2.5×10^9 , 5×10^9 and 1×10^{10} PFU offered significant survival advantages compared to PBS-treated animals, but there was no significant survival advantage imparted by using 5×10^9 and 1×10^{10} compared to the use of 2.5×10^9 . When the dose was further increased to 2×10^{10} PFU, survival was actually reduced compared to 2.5×10^9 PFU/injection. In order to investigate further the mechanism responsible for a drop in survival at this dose, we investigated IL-12 serum levels and histopathology of the liver three days after treatment (Supplementary Fig 2b and c). These data demonstrate minimal IL-12 release into the blood even at high doses, but severe blood vessel congestion as well as eosinophilic degeneration, apoptosis and necrosis of hepatocytes mediated by both Ad-TD-IL-12 and Ad-TD-nsIL-12. These results suggest that the reduced survival on treatment with the highest dose of Ad-TD-nsIL-12 (2×10^{10} PFU/injection) is highly possibly derived from direct viral toxicity, not IL-12 related toxicity.

Minor comment:

The explanation of different ways to control virus replication is not relevant in the discussion section because this is not the main topic of the manuscript.

Reply: We have revised the manuscript based on this comment.

Reviewer #2 (Remarks to the Author):**Overview:**

Interleukin-12 (IL-12) is a potent anti-tumor immunotherapeutic, yet its toxicities after system delivery can be severe and dose-limiting. The authors have re-designed the IL-12 molecule and expressed it from an oncolytic adenovirus with the goals of limiting systemic toxicities while maintaining its anti-cancer immunostimulatory effects. The IL-12 molecule they designed is unable to be secreted from the cell, and therefore is only released from tumor cells following cell lysis. The oncolytic adenovirus into which the transgene for nsIL-12 was engineered is a gene-deleted adenovirus with previously described modifications combined. The Syrian hamster pancreatic tumor model was utilized for studying in vivo efficacy due to previous publications suggesting that this species supports adenovirus replication and toxicity, initially within the lungs. The authors report that this approach is superior to oncolytic adenovirus expressing standard IL-12.

Methods:

The authors should address the following items:

♣ 1. *Activity of IL-12 in Syrian hamsters: The authors state that WBC proliferation is sufficient to demonstrate activity of human IL-12 in this species. However, IL-12 has numerous, complex and species-specific functions. This endpoint alone is not enough to state all important/relevant IL-12 functions are intact within this species.*

Reply: As suggested, IL-12 is indeed a multifunctional cytokine. There are limitations to our study imposed by the lack of available antibodies that are functional in Syrian hamster models. However, in order to address this concern, we designed qPCR primers to detect the changes in TNF- α and IFN- γ mRNA levels in hamster splenocytes to provide further evidence for the activity of human IL-12 in hamster models besides our original lymphocyte proliferation assay. This has been included in Figure 3 panels j and k.

♣ 2. *Efficacy data: The authors should consider treating tumors at a more advanced stage to better reflect the clinical situation. These animals were treated shortly after tumor cell implantation when tumors were not yet established and/or confirmed to be metastatic.*

Reply: In response to this comment, and to show translational relevance of our models, we show in our revised manuscript that four or six days (model-dependent) is adequate for replication of late-stage disease. At six days after intrapancreatic injection, Hap-T1 tumors reached 6-7mm in diameter, and some tiny metastatic tumors were found in the liver (Supplementary Fig. 1). At four days after intraperitoneal injection of SHPC6 cells, the abdominal cavity contained a large amount of serosanguinous ascites fluid, multiple oval-shaped nodules had localized and grown in the mesentery adjacent to the pancreas and spleen (Supplementary Fig 1).

All of these pathological conditions show intraperitoneal and intrapancreatic tumors are in advanced stages and thus do represent the most commonly diagnosed advanced stage disease.

We have clarified this within the relevant text sections:

Ad-TD-nsIL-12 is an effective and non-toxic anti-tumor agent after systemic delivery for treatment of peritoneally disseminated PaCa
And

Ad-TD-nsIL-12 retains superior antitumor efficacy in a clinically relevant orthotopic pancreatic cancer model

We have also added supplementary figure 1 to show images of tumours/pathology in these models after such short timeframes.

♣ 3. *Route of administration: The authors should test IV therapy given that pancreatic cancer is frequently systemic and metastasized after diagnosis. In addition, pancreatic tumors are extremely difficult to inject directly clinically given the high intratumoral pressure and fibrosis.*

Reply: We completely agree with this reviewer's comment that as pancreatic cancer is frequently systemic and metastasized after diagnosis, IV injection would be ideal for this aggressive disease. However, it has been demonstrated that Ad5 adenovirus is not suitable for intravenous injection. Adenovirus half-life was less than two min after a single intravenous injection of concentrated virus, which was quickly cleared by liver Kupfer cells. Coating with PEG ('PEGylation') reduced the clearance rate of adenovirus but also reduced infectivity (Alemany R et al. J Gen Virol 2000, 17:761-70). In addition the high titer of adenovirus neutralizing antibody in patient blood also limits the IV injection efficiency of oncolytic adenoviruses. Oncolytic adenoviruses currently in clinical use or development are usually delivered via intratumoral or intraperitoneal injection. Therefore in our present study we chose intratumoral or intraperitoneal injection instead of IV injection.

It is true that pancreatic tumors are extremely difficult to inject directly clinically given the high intratumoral pressure and fibrosis. We injected the viruses through a single central tumor puncture site, and three to four needle tracks were made radially from the center while virus was injected as the needle was withdrawn. In clinical treatment, the injections are performed under the guidance of endoscopic ultrasound or CT.

♣ 4. *Cancer cell selectivity in vitro: This is assessed in a single human lung cancer line and one normal cell type. This data on selectivity and efficacy should be expanded with more*

cancer lines (including pancreatic) and normal cell sources. Wildtype and gene deletion Adeno controls should also be evaluated in parallel.

Reply: Following this insightful comment, we detected the selectivity and efficacy of wild-type and novel oncolytic adenoviruses in more human cancer cells, including pancreatic cancer, breast cancer, ovarian cancer and prostate cancer (Figure 1). Given the key normal organs infected by adenovirus (lung and liver), we have also included a comparison between Ad5 and Ad-TD-LUC in normal human bronchial epithelial cells and normal Syrian hamster hepatocytes (Figure 1 b & c) that demonstrates improved tumour selectivity of our modified virus. Furthermore we have now included *in vivo* analysis of tumor, lung and liver tissue after treatment with Ad5 versus Ad-TD-LUC (Fig. 1l) or Ad-TD-IL12 versus Ad-TD-nsIL12 (Fig 6e). Both of these figures confirm the lack of infection/replication in normal tissue (liver/lung) by our Ad-TD viruses and improved tumour cell infection/replication of Ad-TD-LUC compared to Ad5.

♣ 5. *Cancer selectivity assessed in vivo: This study assessed E1A gene copies per unit tissue. However, it is possible that replication was incomplete/inefficient in hamster tissue, and therefore the infectious virus titers may be significantly lower. The dosing frequency (every two days for 10 days total) may also result in virus genomes being present despite inefficient replication. The authors should assess plaque-forming units (pfu) in addition to E1A gene copies. In addition, other normal tissues should be assessed (eg proliferating tissues which may be more supportive of viral replication). Wildtype and gene deletion Adeno controls should also be evaluated in parallel. Finally, replication and selectivity of the Ad-nsIL-12 virus should be assessed in vivo.*

Reply: We did assess plaque-forming units (pfu) in addition to E1A gene copies, and the result was shown in Figure 6d. We also detected the E1A gene copies of Ad-TD-nsIL-12 in tumor, liver and lung (Fig 6e). Control viruses (Ad5 and AD-TD-LUC) results are shown as Figure 1L, and IL-12 and ns-IL12 modified viruses shown in Figure 6e. These results

demonstrate that our modified virus (AD-TD-LUC) with or without IL-12 is detected at very low limits (on the border of assay sensitivity) compared to WT Ad5, which was detected at significantly higher levels in normal tissues, demonstrating *in vivo* tumor selectivity of our modified virus.

♣ 6. *Syrian hamster cell line infection and replication: These data should be shown alongside data on replication of both Adeno gene-deletion mutants (as was done), wildtype adenovirus control(s), and all viruses in human pancreatic cancer cell lines. This data is necessary to interpret the replication efficiency of these gene-deletion adenovirus constructs in hamster cell lines.*

Reply: We have now incorporated the replication data for wild-type adenovirus and Adeno gene-deletion mutants in hamster and human cancer cells. These are shown in Figures 1 and 2.

Reviewer #3 (Remarks to the Author):

In this manuscript, Wang et al. report success in treating a Syrian hamster model of pancreatic cancer with an altered oncolytic adenovirus delivering modified human IL-12. Overall, the study builds on the generally accepted idea that IL-12 delivery could improve anti-tumor responses due to many of its immune-related actions, particularly if its related toxicity could be reduced. Indeed, this is seemingly what this group has accomplished. Because of this and the fact that this group used a human IL-12, which could be quickly translated to clinical trial, this manuscript should be published. While much of the work presented is well rationalized and executed, a number of issues should be addressed before this manuscript is suitable for publication:

Major (in order of importance, with most important being listed first):

1) The comparison of Figure 7a with modified IL-12-treated but CD4-depleted or CD8-depleted hamsters to IL-12 treated non-depleted hamsters is not the correct control. The question remains what is the growth of tumors in untreated but CD4-depleted and CD8-depleted hamsters. What if the tumor growth curves are the same in IL-12-treated and untreated hamsters that have been CD4-depleted or CD8-depleted. Then CD4s or CD8s would seemingly not play any role (but that cannot be determined without the untreated CD4-depleted and untreated CD8-depleted controls).

Reply: We appreciate this suggestion and have performed the animal study based on the comment. The result has been added to Figure 7a and 7b. Tumor growth equivalent to PBS groups was noted in hamsters treated with CD4 or CD3 depletion antibodies alone. This suggests that depletion of CD3 or CD4 T cells does not affect tumor growth of Syrian hamster HPD1NR pancreatic tumor. Depletion of CD3 cells had a significant impact on treatment efficacy in the Ad-TD-nsIL-12 treated animals.

2) The conclusions from Figure 8 are too far-reaching. There is a statement made that “these observations suggest that the improved safety associated with nsIL-12 delivery is due to reduction, but not abrogation of the inflammatory responses generate by IL-12.” While I agree that reduced inflammatory responses generated by IL-12 are shown, there is no indication that this has anything to do with improve safety. These two findings may be unrelated. Some indication that putting inflammation back (via CPG ODN or poly IC) in the context of nsIL-12 reproduces the lack of efficacy of normal IL-12, would “suggest” the claim made here. Otherwise, this claim should be removed.

Reply: The reviewer is correct here and we have revised the statement to tone it down. This is an area that we are currently attempting to investigate further and we hope to publish these results at a later date.

3) Inconsistent statistical analyses: some figures seemingly have no statistical analyses (including figure 2, figure 3, panels c, d, and f of figure 6, figure 8) although seemingly differences are claimed to exist among groups within those figures and their panels.

Reply: Apologies for this error, we have included the statistical analyses for all figures based on the comments.

4) In Figure 5f, an important rechallenge experiment (40 days later) is conducted to demonstrate long-term efficacy, and the proper rechallenge control of naïve hamsters being injected is mentioned in the text. However, in the figure it seems that only such naïve hamsters were used in the original challenge and none were used after the day 40 rechallenge (to show how much different equally age-matched naïve hamsters would respond). Another (maybe even more pertinent) manner in which to ask this question is: if hamsters were challenged with this tumor, then the tumor was excised (without the mice ever receiving the modified IL-12), would they demonstrate response to the rechallenge shown in Figure 5f, or would they essentially appear naïve?

Reply: In figure 5f, 10 hamsters were intraperitoneally injected with 1×10^7 SHPC6 cells, then treated with Ad-TD-nsIL-12. 40 days later, the Ad-TD-nsIL-12 treated group and same age naïve hamsters were re-challenged with 1×10^7 SHPC6 cells. We have revised the figure to make it clearer. Regarding the reviewer's suggestion that the tumour is left untreated, excised and then the animal rechallenged, we agree that this would be an effective way of analysing this phenomenon, however SHPC6 cells are rapidly disseminated, thus tumour excision is impossible.

Minor: 1) Line 41 states that tumor-induced suppression is "the" major mechanism by which tumors evade immune-mediated detection. While it is certainly "a" mechanism, it may not be "the" mechanism since it has not been shown to be more "major" than the lack of

immune infiltration (i.e., cold tumors), neoantigen sequestration from the immune system, and immune effector exhaustion (against which checkpoint inhibitors have been successful in clinical trials and as first line treatment regimens). This statement should be revised.

Reply: We have revised the manuscript based on this comment.

REVIEWERS' COMMENTS:

Reviewer #1 (Remarks to the Author):

The authors have made a remarkable effort to improve the manuscript based on the reviewer's recommendations. Most technical points raised in the previous round of review have been satisfactorily addressed. This work clearly shows that the therapeutic index of an oncolytic adenovirus expressing IL-12 can be increased by deleting the signal peptide of the cytokine. Until deeper analysis of the mechanism of action is available, the most logical explanation is that this crippled IL-12 version confers a positive balance between toxicity and efficacy to the vector. Combination with other strategies will be needed to achieve optimal results in the most relevant tumor model.

Reviewer #2 (Remarks to the Author):

The authors have clearly answered each question posed in the previous review. Additional data has been added on cancer vs normal cell selectivity, and on the extent of tumor progression within these animal tumor models at the time of treatment.

Reviewer #3 (Remarks to the Author):

The author responses and revised manuscript has adequately addressed all of my previous comments.